# Calibrating Verbalized Confidence with Self-Generated Distractors

**Victor Wang**        **Elias Stengel-Eskin**

The University of Texas at Austin

## Abstract

Calibrated confidence estimates are necessary for large language model (LLM) outputs to be trusted by human users. While LLMs can express their confidence in human-interpretable ways, verbalized LLM-generated confidence scores have empirically been found to be miscalibrated, reporting high confidence on instances with low accuracy and thereby harming trust and safety. We hypothesize that this overconfidence often stems from a given LLM's heightened *suggestibility* when faced with claims that it encodes little information about; we empirically validate this hypothesis, finding more suggestibility on lower-accuracy claims. Building on this finding, we introduce Distractor-Normalized Coherence (DiNCo), which estimates and accounts for an LLM's suggestibility bias by having the model verbalize its confidence independently across several self-generated distractors (i.e. alternative claims), and normalizes by the total verbalized confidence. To further improve calibration, we leverage generator-validator disagreement, augmenting normalized validator confidence with a consistency-based estimate of generator confidence. Here, we frame the popular approach of self-consistency as leveraging coherence across sampled generations, and normalized verbalized confidence as leveraging coherence across validations on incompatible claims, allowing us to integrate these complementary dimensions of coherence into DiNCo. Moreover, our analysis shows that DiNCo provides less saturated – and therefore more usable – confidence estimates, and that further sampling alone cannot close the gap between DiNCo and baselines, with DiNCo at 10 inference calls outperforming self-consistency at 100.[1]

## 1 Introduction

LLMs encode a vast amount of knowledge in their parameters, demonstrating superhuman performance on knowledge-intensive benchmarks (Comanici et al., 2025; OpenAI, 2023). Users often rely on information obtained from these models to make important decisions, but this information is not always accurate. Thus, we seek to qualify LLM responses with confidence estimates that are *calibrated*, i.e. match the probability of correctness. Users and agentic frameworks often use LLMs off the shelf without task-specific or model-specific tuning (Manakul et al., 2023; Geng et al., 2024; Feng et al., 2024; Shorinwa et al., 2025), motivating the development of confidence estimation methods that work in off-the-shelf settings – both gray-box settings with logit access, and black-box settings with only textual input and output.

In these settings, verbalized confidence is a simple and commonly-used approach that prompts the model to report its confidence in an answer (Lin et al., 2022; Xiong et al., 2024; Wei et al., 2024). For brevity, we use *verbalized confidence* as a blanket term for (1) asking the model to decode a numerical confidence in text like *"80%"* (Tian et al., 2023) and (2) asking the model whether an answer is correct and taking the token probability $P(\text{True})$ (Kadavath et al., 2022). Verbalized confidence is appealing for several reasons, including that it resembles one way humans express confidence, making it easy to interpret and integrate into decision-theoretic frameworks (Sun et al., 2025; Steyvers et al., 2025). However, verbalized confidence has several drawbacks. First, it empirically tends to exhibit overconfidence (Tian et al., 2023; Xiong et al., 2024; Wei et al., 2024; Xu et al., 2025);

---

[1]Code: https://github.com/victorwang37/dinco

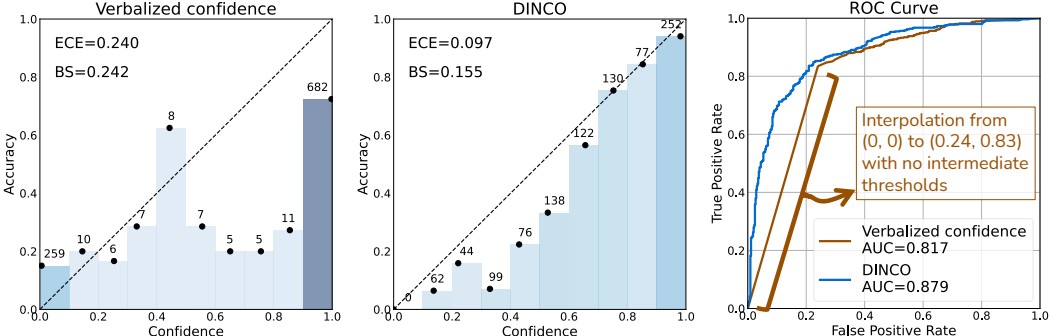

Figure 1: Calibration metrics (Expected Calibration Error ↓, Brier score ↓, area under the ROC curve ↑; see Appendix B.1) with Qwen3-8B on TriviaQA using P(True) as verbalized confidence. **(Left)** Verbalized confidence is saturated at high confidence, despite a much lower accuracy. For each bar, we label the number of instances whose confidence falls in the interval and we darken larger bins. **(Center)** DINCO normalizes by the total confidence over candidate answers, relieving saturation and improving calibration. **(Right)** Since verbalized confidence is saturated at high confidence, it is unable to achieve an acceptable true positive rate (TPR) without incurring a significant false positive rate (FPR) of 0.24. In other words, no rejection threshold can be chosen to reject a high proportion of false claims. Meanwhile, DINCO enjoys better granularity, ranking positives above negatives even among instances with a verbalized confidence of 1.

Fig. 1 (left) shows that verbalized confidence scores generally outstrip average accuracy within a confidence bin.

We highlight a second underexplored factor that makes verbalized confidence suboptimal: **confidence saturation**, wherein the model's reported scores tend to fall into a few bins, making them uninformative. While this might still lead to an acceptable calibration error, it results in "jumpy" curves, as in Fig. 1 (right), where no confidence threshold that accepts at least one claim can avoid accepting a substantial proportion of false claims. To address these shortcomings, we introduce DINCO, which leverages incoherence in verbalized confidence across related claims to detect over-confidence. DINCO is motivated by the intuition that incoherent confidence scores, e.g., a high verbalized confidence in an answer to a question when other distinct answers also have high verbalized confidence, should not be taken at face value. In other words, we should discount high confidence if it does not follow rational coherence norms (Hofweber et al., 2024).

To explain and correct for this kind of incoherence, we first define the notion of suggestibility. Some studies indicate that when LLMs are epistemically uncertain, they tend to rely on their context to resolve the uncertainty (Yadkori et al., 2024; Ahdritz et al., 2024), i.e., the confidence on a claim increases *because* it is in the context. We refer to this phenomenon as *suggestibility* and hypothesize that it contributes to the model's assignment of high confidence to claims it can neither support nor refute. We introduce this hypothesis in Section 2.1 and provide empirical support in Section 2.2. To account for this suggestibility bias, we propose a method for calibrating verbalized confidence that normalizes by the total confidence over self-generated distractors (i.e. alternative claims). We generate minimal pair distractors using beam search when available, or by directly prompting the model for distractors in the black-box setting. Crucially, we use an off-the-shelf NLI model to downweight distractors that are similar to other distractors or that do not contradict the main claim.

The approach above for normalizing verbalized confidence with distractors aims to leverage coherence within claim validation, but overlooks another relevant facet of coherence in LLMs. In particular, coherence among sampled generations is correlated with correctness, an observation leveraged by the popular approach of self-consistency (Xiong et al., 2024). Thus, inspired by prior findings on generator-validator disagreement (Li et al., 2024), we integrate these complementary dimensions of coherence into DINCO. Specifically, we use distractor generation and NLI reweighting to estimate and enforce coherence across validations of related claims (e.g. not accepting contradictory claims), while using self-consistency to quantify coherence across sampled generations, upweighting more commonly generated claims.

We test our method on open-source and closed-source models, applied to short-form (TriviaQA and SimpleQA) (Joshi et al., 2017; Wei et al., 2024) and long-form (FactScore; Min et al., 2023) generation domains. DiNCo improves ECE over the best baseline by an average of 0.077, 0.092, and 0.055, respectively (note that the best baseline differs between the short-form and long-form settings). DiNCo effectively extends to long-form biography generation, where it improves Pearson and Spearman correlation with passage-level FactScore over the best baseline by an average of 0.072 and 0.074, respectively. Further analysis shows that DiNCo relieves confidence saturation, and that simply scaling up self-consistency (the strongest baseline overall) does not suffice to match the calibration of DiNCo.

## 2 DISTRACTOR-NORMALIZED COHERENCE (DiNCo)

We begin with a motivating hypothesis, supported with preliminary evidence. Then we present the details of our method, illustrated in Fig. 2.

### 2.1 BACKGROUND AND MOTIVATION

Let $\mathcal{C}$ be the set of claims with a binary truth value. We denote the truth value of a claim $c \in \mathcal{C}$ as $v(c) \in \{0, 1\}$. A confidence estimation method is a function $f : \mathcal{C} \to [0, 1]$, which is *calibrated* if it correctly predicts the probability of truth. Verbalized confidence is an approach that prompts an LLM to output its confidence $f^{\text{VC}}(c)$ in a claim $c$.[2]

For a topic that the model knows little about, it may be willing to adopt the information presented in its context as its prior (Yadkori et al., 2024; Ahdritz et al., 2024), a phenomenon we refer to as *suggestibility*. In other words, the very act of presenting a claim for the model to report its confidence on can bias the reported confidence. For example, if the model does not know who Kang Ji-hwan is, it may assign 60% confidence to both the claim *"Kang Ji-hwan was born in 1980."* and the claim *"Kang Ji-hwan was born in 1990."*, even though this seemingly violates coherence norms (since the claims are mutually exclusive). Nonetheless, this behavior may not be strictly irrational, as each confidence estimate is conditioned on different information, namely the fact that the respective claim was verbalized in the user prompt. In other words, the very fact that the claim has been provided in the input might lend credence to the claim. We further discuss the connection between this notion, LLM sycophancy, and human suggestibility in Appendix A.1.

We seek to correct for this bias caused by the model's suggestibility when presented with claims it knows little about. Let $f^{\text{VC}}$ be the verbalized confidence, and let $f^{\text{lat}}$ be the latent, inaccessible model confidence. We model the bias as a multiplicative scalar $\beta(c)$, which depends on the claim $c$ because the model has varying degrees of uncertainty for different topics: $f^{\text{VC}}(c) = \beta(c)f^{\text{lat}}(c)$. To approximate $f^{\text{lat}}$, we make the assumption that the biases for logically related (e.g., equivalent or contradictory) claims are approximately equal, since they rely on a shared, localized set of knowledge. Let $C \subset \mathcal{C}$ be a set of mutually exclusive and exhaustive claims, e.g. claims for the year a person was born in. Since the claims in $C$ are logically related, we assume that $\beta(c)$ is roughly the same for all $c \in C$ and so there is a scalar $\beta(C)$ with $\beta(C) \approx \beta(c)$ for all $c \in C$. Assuming the latent confidence $f^{\text{lat}}$ is probabilistically coherent,

$$1 = \sum_{c \in C} f^{\text{lat}}(c) = \sum_{c \in C} \frac{f^{\text{VC}}(c)}{\beta(c)} \approx \sum_{c \in C} \frac{f^{\text{VC}}(c)}{\beta(C)}. \tag{1}$$

Thus, we can approximate $\beta(C)$ and then $f^{\text{lat}}$:

$$\beta(C) \approx \sum_{c \in C} f^{\text{VC}}(c), \qquad f^{\text{NVC}}(c) = \frac{f^{\text{VC}}(c)}{\beta(C)} \approx f^{\text{lat}}(c) \tag{2}$$

In practice, we set $\beta(C) \leftarrow \max(1, \beta(C))$ to account for the case where $C$ fails to contain a true claim, or more precisely, a claim that the model believes to be true.

---

[2]We talk about what the model knows, believes, has confidence in, etc. as short-hand notation for the latent ability to produce language similar to that which a human would to demonstrate such knowledge, etc. (Piantadosi & Hill, 2022; West et al., 2024; Hofweber et al., 2024).

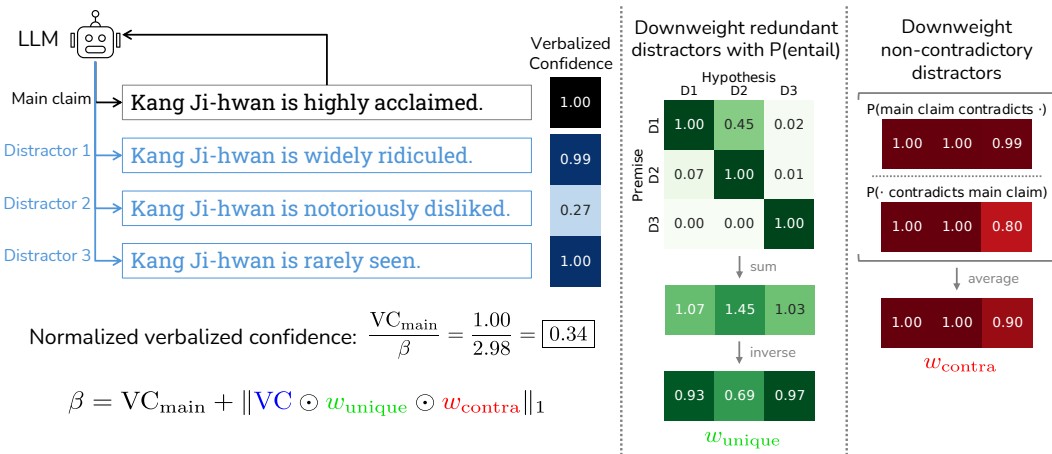

Figure 2: Normalizing verbalized confidence with DINCO. **(Left)** The LLM generates a claim along with several distractors and reports its confidences on them independently. To calibrate the main claim's confidence, we divide it by $\beta$, the sum over each distractor's confidence, weighted by uniqueness **(center)** and counterfactuality **(right)**. We seek distractors that are minimal pairs with the main claim, i.e. similar statements that likely contradict the main claim. The distractors shown here are real examples generated by an LLM, and while they happen to share a lexical structure with the main claim, we do not prescribe a precise form for generating distractors; see Section 2.3 for method details, Appendix B.2 for prompts, and Appendix C.1 for more examples and analysis.

## 2.2 PRELIMINARY STUDY

To empirically support our motivation above, we plot the distributions of the total confidence ($\beta(C)$ in Eq. 2) over correctly and incorrectly answered questions. In this preliminary study, we take correctness as a proxy for epistemic certainty, although in reality LLMs may still be uncertain about questions they answered correctly. In other words, we treat correct and incorrect instances as epistemically certain and uncertain instances, respectively. Thus, our hypothesis predicts that on incorrect instances, the model would be *more suggestible* and assign high confidence to more answers, leading to *higher* total confidences. On the other hand, if the model is calibrated (in particular, on uncertain instances, it exhibits epistemic humility, i.e. recognizing its lack of knowledge), then the total confidence would tend to be around 1 and 0 for correct and incorrect instances, respectively, so the total confidence would be *lower* on incorrect instances. If the model is confident in its incorrect answers (e.g. due to misconceptions), we would expect *similar* behavior between correct and incorrect instances.

**Experimental Setup.** We use TriviaQA (Joshi et al., 2017), a dataset evaluating real-world knowledge with short-form answers. We sample 1000 questions from the validation split of the `rc.nocontext` subset. We use Qwen3-8B (Yang et al., 2025)[3] and take verbalized confidence using P(True) (Kadavath et al., 2022), computed as $P(\text{Yes})/(P(\text{Yes}) + P(\text{No}))$ when asking the model whether a given answer is correct (see Appendix B.2 for prompts).

For each question, we generate 10 claims, so the total confidence is a number from 0 to 10. In Section 2.3, we specify how the distractors are generated and how we avoid overcounting redundant answers in the total confidence (Eq. 4). The total confidence $\beta(C)$ computed here matches how the NVC method and the NVC component of DINCO (but with fewer distractors) will be computed for our main experiments in Section 3.

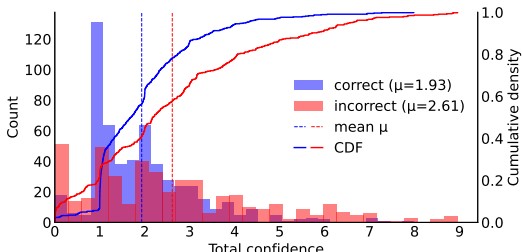

Figure 3: Total confidence for correctly and incorrectly answered questions.

**Results.** In Fig. 3, we first observe that some incorrectly answered questions have a to-

---

[3]Throughout, we use the instruction-tuned versions of Qwen3 models.

tal confidence near 0, suggesting some epistemic humility. Even so, the incorrect distribution is heavy-tailed, resulting in a higher mean and median than the correct distribution. These results are consistent with our hypothesis that LLMs are more prone to accepting claims that they are epistemically uncertain about. To explain why the model experiences suggestibility even on correct instances (shown by the substantial proportion of correct instances with a total confidence greater than 1), we note that educated guesses can often be correct while still bearing epistemic uncertainty. Despite our usage of correctness as a noisy proxy of epistemic certainty, we identify the clear trend that the model tends to report higher confidence on claims for incorrectly answered questions. We verify the same trend on SimpleQA (Wei et al., 2024) and biography generation (Min et al., 2023) in Appendix D.1.

## 2.3 METHOD

Our preliminary result (Fig. 3) shows that LLMs can produce incoherent probability judgments (i.e. the total confidence $\beta(C)$ exceeds 1), especially when epistemically uncertain, suggesting a need to estimate and correct for this bias. As proposed in Section 2.1, we normalize verbalized confidence by the total confidence $\beta(C)$ over a distractor set $C$. We now describe in detail how we generate these distractors, and how we account for redundancy among distractors, a process illustrated in Fig. 2. Finally, we discuss how the phenomenon of generator-validator disagreement motivates incorporating self-consistency into DiNCo.

**Distractor generation.** The distractor set should contain enough plausible distractors to avoid underestimating the normalization factor ($\beta(C)$ in Eq. 2), while remaining small enough to be tractably computed. Thus, we frame the problem of choosing the optimal distractor set containing an original claim $c_0$ as maximizing the total acceptance probability $\sum_{c \in C} f^{\mathrm{VC}}(c)$ subject to a size constraint $|C| \leq K$; shortly we address relaxing the requirement of mutual exclusivity. Unfortunately, the validation probability $f^{\mathrm{VC}}(c)$ can only be elicited on a per-claim basis, leading to intractable sample complexity. Motivated by the intuition that an LLM tends to generate claims that it would find plausible during validation, as a proxy for the set of claims with high verbalized confidence, we use the set of claims with high generation probability (under an appropriate prompt, explained next).

To encourage mutual exclusivity among claims, we set up the claims to be minimal pairs (Fig. 2 left). For a given short-form question, we can simply sample many answers, but independent sampling is inefficient as it overrepresents high probability generations (Gekhman et al., 2025), limiting the number of unique distractors. Instead, we use beam search (Sutskever et al., 2014) when available to efficiently identify unique sequences that have high probability mass coverage. For API-access models, we use top token probabilities if available to implement a pseudo-beam search (see Appendix C.2 for details). Otherwise in the black-box setting, we directly prompt the model to generate a list of candidate answers (see Appendix B.2 for prompts). For long-form QA, we follow Min et al. (2023) in decomposing a long generation into claims. We separately prompt the model to generate one distractor for a given claim, and again use beam search to create multiple distractors. We provide examples and analysis of the distractors generated with each of these methods in Appendix C.1.

**Addressing Claim Redundancy.** Although the heuristic of generating minimal pairs encourages mutually exclusive claims, we have little guarantee of this mutual exclusivity (1) within the distractor set $C$ and (2) between distractors and the original claim. Assuming such mutual exclusivity when there are actually redundant claims can lead to overcounting in the normalization factor $\beta(C)$. Thus, we use an NLI model to quantify entailment and contradiction relationships between claims.[4] We address (1) and (2) with $w_{\mathrm{unique}}$ and $w_{\mathrm{contra}}$, respectively (Fig. 2):

$$w_{\mathrm{unique}}(c) = \frac{1}{\sum_{c' \in C} P(\mathrm{entail} \mid c', c)}, \quad w_{\mathrm{contra}}(c) = \frac{P(\mathrm{contra} \mid c_0, c) + P(\mathrm{contra} \mid c, c_0)}{2} \quad (3)$$

Intuitively, $w_{\mathrm{unique}}$ downweights a claim if it is entailed by other claims, and $w_{\mathrm{contra}}$ downweights a claim if it is not contradictory with the main claim. In the simplified setting where claims are either fully equivalent or fully contradictory, $w_{\mathrm{unique}}$ for a claim is the reciprocal of the size of its equivalence class, so we have invariance to claim duplication. Similarly, $w_{\mathrm{contra}}$ grants invariance to

---

[4]Access to an NLI model poses only a minimal departure from the zero-resource setting, since NLI is a generic task for which there are off-the-shelf models. NLI is a subset of the tasks that LLMs are capable of, so the usage of a separate NLI model is motivated merely by efficiency (Kuhn et al., 2023; Lin et al., 2024).

including the original claim as a distractor. Beyond this simplified setting, using continuous weights allows us to model partial entailment or contradiction, such as for the claims *"Kang Ji-hwan is widely ridiculed."* and *"Kang Ji-hwan is notoriously disliked."* in Fig. 2.

We now normalize the verbalized confidence of the original claim as

$$f^{\text{NVC}}(c_0) = \frac{f^{\text{VC}}(c_0)}{\beta(C)}, \quad \beta(C) = \max\left(1, f^{\text{VC}}(c_0) + \sum_{c \in C} f^{\text{VC}}(c) \cdot w_{\text{unique}}(c) \cdot w_{\text{contra}}(c)\right),$$
(4)

generalizing the mutually exclusive case in Eq. 2. The maximization with 1 allows for defaulting back to the vanilla verbalized confidence in the case where $C$ fails to contain claims that the model considers plausible.

**Combining Coherence within Generation and Validation.** Our approach so far (summarized in Fig. 2) for normalizing verbalized confidence across distractors focuses on coherence within claim *validation*. Previous studies disagree on the question of whether models are better at the discriminative or generative counterparts of a given task (West et al., 2024; Gekhman et al., 2025), but generally agree on the presence of generator-validator inconsistency (Li et al., 2024), wherein a model may produce inconsistent results between the generation and validation stages. Indeed, we find that in the preliminary study setting in Section 2.2, the answer with the highest generation probability and the answer with the highest validation probability (over 10 answers obtained from beam search) agree[5] on only 592 out of 1000 questions.

We integrate these complementary aspects of generation and validation into DINCO. In particular, we draw on a distributional view of the generator-validator gap (Rodriguez et al., 2025), in which the generation probability distribution over candidate answers (which we approximate with self-consistency sampling, $f^{\text{SC}}$) is distinct from the validation probability distribution over them (which we approximate with normalized verbalized confidence, $f^{\text{NVC}}$). DINCO thus incorporates confidence (i.e. probability mass) in both the generator and validator distributions: $f^{\text{DINCO}}(c) = \frac{1}{2}f^{\text{SC}}(c) + \frac{1}{2}f^{\text{NVC}}(c)$. We leave a full description of the self-consistency component to Appendix C.3.

## 3 EXPERIMENTS

### 3.1 EXPERIMENTAL SETUP

**Short-form Datasets.** Short-form QA serves as a testbed for evaluating factuality as well as calibration because of its tractable evaluation and adjustable difficulty. The task is relevant in practice because it assesses models' ability to respond to information-seeking users. TriviaQA contains trivia questions requiring diverse world knowledge (Joshi et al., 2017). SimpleQA similarly contains short, fact-seeking questions, curated with the criterion of challenging frontier models (Wei et al., 2024). We sample 1000 questions from each dataset.[6] In Appendix D.6, we evaluate on BioASQ (Krithara et al., 2023), a dataset demanding biomedical expertise, to show generalization to other domains. We use LLM-as-a-judge to evaluate binary correctness, following best practices for robust evaluation (Wei et al., 2024); in Appendix B.3 we confirm high human agreement.

**Long-form Datasets.** While short-form settings are appealing for their simple evaluation, many real-world tasks require longer generations, for which calibrated confidence estimation remains critical. The long-form setting comes with the evaluation challenge that responses generally contain both correct and incorrect parts, complicating the assignment of a single correctness score. In our experiments, we evaluate long-form calibration on biography generation using FactScore (Min et al., 2023). FactScore decomposes a generated biography into atomic claims and verifies each claim against Wikipedia (see Appendix B.4 for an example), thus enabling us to evaluate calibration at the claim level. We use the labeled subset containing 183 entities from Min et al. (2023).

---

[5] We consider $c$ and $c'$ equivalent answers to question $q$ if $\frac{1}{2}P(\text{entail} \mid c, c'; q) + \frac{1}{2}P(\text{entail} \mid c', c; q) > 0.9$.

[6] We use the validation split of the `rc.nocontext` subset for TriviaQA. On SimpleQA, Gemini-2.5-Flash produced a refusal error with no output on 86 questions, so we exclude them from all experiments.

**Models.** Since TriviaQA and SimpleQA are adequately challenging for smaller and larger models, respectively, we focus their evaluation accordingly. TriviaQA is largely solved by frontier models (Wei et al., 2024), and SimpleQA is too difficult for smaller models.[7] On TriviaQA, we use popular open-source models: Qwen3-32B, Qwen3-8B, and Qwen3-1.7B (Yang et al., 2025), Llama-3.2-3B-Instruct (Dubey et al., 2024), and Gemma-3-4B-IT (Team et al., 2025). On SimpleQA, we use popular frontier models: GPT-4.1 (2025-04-14; OpenAI, 2025) and Gemini-2.5-Flash (Comanici et al., 2025). For SimpleQA evaluated on frontier models, we also consider the black-box setting where no logit access is assumed; here, rather than using pseudo-beam search to generate distractors, we prompt the model directly to generate diverse distractors. Moreover, we replace P(True) with verbalized numerical confidence to forgo logit access. For the long-form task of biography generation, we limit our evaluation to Qwen3-8B and Gemma-3-4B-IT, due to the cost of FactScore evaluation with GPT-4.1. We use the NLI model `DeBERTa-v3-base-mnli-fever-anli` (He et al., 2021) for our methods and self-consistency; we confirm robustness to the choice of the NLI model in Appendix D.3.

**Evaluation Metrics.** We evaluate Expected Calibration Error (**ECE** ↓; Naeini et al., 2015) with 10 bins, Brier score (**BS** ↓; Brier, 1950), and area under the ROC curve (**AUC** ↑; Hanley & McNeil, 1982). See Appendix B.1 for descriptions. Like accuracy, these calibration metrics are on a scale from 0 to 1, meaning that e.g., an improvement of 0.05 is substantial, as it corresponds to a 5% absolute improvement (Tian et al., 2023; Xiong et al., 2024); we report all improvements in absolute rather than relative terms. For biography generation, we also evaluate Pearson and Spearman correlation between average claim-level confidence and passage-level FactScore (without length penalty), where the latter measures the proportion of claims that are correct.

**Baselines.** Following past work in zero-resource calibration, we compare against training-free methods that produce probabilities without post-hoc calibration (Tian et al., 2023; Xiong et al., 2024; Steyvers et al., 2025). We provide prompts for our methods and baselines in Appendix B.2. Verbalized confidence (**VC**; P(True) from Kadavath et al., 2022) asks the model whether its answer is correct and computes $P(\text{Yes})/(P(\text{Yes}) + P(\text{No}))$. It is straightforward to replace P(True) with verbalized numerical confidence; in Appendix D.2 we show that the latter similarly benefits from our methods, showing robustness to the format of verbalized confidence. Top-$K$ prompting ($K$-**VC**; Verb. 1S from Tian et al., 2023) prompts the model to provide its top $K$ guesses along with verbalized numerical confidences. For our black-box setting, we use the candidate answers generated using the $K$-VC prompt as distractors, but we discard the verbalized confidences contained in the same generation, and instead separately collect verbalized numerical confidence on each distractor independently. Maximum sequence probability (**MSP**; Fadeeva et al., 2023) is the LLM's probability of generating its answer. Self-consistency (**SC**; Xiong et al., 2024) samples several answers (we use temperature 1) and computes the proportion that match the main answer. Following Kuhn et al. (2023), we use an NLI model to determine semantic equivalence when grouping together answers for SC.[5] For biography generation, we modify self-consistency to sample several biographies and measure entailment of each claim (Zhang et al., 2024a). **SC-VC** (Xiong et al., 2024) is SC weighted by verbalized confidence (we use P(True) following Taubenfeld et al., 2025).

**Inference Budget.** As our methods and several baselines ($K$-VC, SC, SC-VC) operate on a variable inference-time budget, we control this budget at $K = 10$. For DINCO, we use 5 samples for self-consistency and 5 distractors for normalized verbalized confidence; we show robustness to the budget split for various budgets in Appendix D.4 and thus recommend an equal split for simplicity.

## 3.2 RESULTS

**Short-form QA.** Tables 1 and 2 report that on TriviaQA and SimpleQA, DINCO outperforms the best baseline, MSP, by an average ECE of 0.077 and 0.092, respectively. While MSP is a competitive baseline (e.g. AUC 0.800 surpasses DINCO at 0.786 on SimpleQA with GPT-4.1), this often does not hold across metrics (e.g. MSP has an ECE of 0.263 in the same setting, heavily underperforming DINCO at 0.089) or across settings (e.g. MSP underperforms DINCO on AUC in TriviaQA by an average of 0.062). Most importantly, the effectiveness of MSP relies on answers having a canonical form, restricting its usage to multiple-choice or short-form questions and preventing its generalization to long-form settings (Farquhar et al., 2024). We highlight that NVC outperforms SC (e.g. by

---

[7] https://www.kaggle.com/benchmarks/openai/simpleqa

Table 1: TriviaQA results. We evaluate Expected Calibration Error (ECE), Brier score (BS), and area under the ROC curve (AUC). In each column, we bold the best result and underline results not significantly worse under a paired test ($\alpha = 0.05$; see Appendix B.5 for tests). For readability of this table, we leave Qwen3-32B results to Appendix D.5, where we verify that the effectiveness of DINCO extends to larger scales of open-source models.

| Method | Qwen3-8B | | | Qwen3-1.7B | | | Llama-3.2-3B-Instruct | | | Gemma-3-4B-IT | | |
|---|---|---|---|---|---|---|---|---|---|---|---|---|
| | ECE↓ | BS↓ | AUC↑ | ECE↓ | BS↓ | AUC↑ | ECE↓ | BS↓ | AUC↑ | ECE↓ | BS↓ | AUC↑ |
| VC (Kadavath et al., 2022) | 0.240 | 0.242 | 0.817 | 0.387 | 0.383 | 0.720 | 0.189 | 0.208 | 0.826 | 0.300 | 0.299 | 0.702 |
| K-VC (Tian et al., 2023) | 0.341 | 0.348 | 0.604 | 0.538 | 0.524 | 0.596 | 0.146 | 0.228 | 0.678 | 0.254 | 0.262 | 0.786 |
| MSP (Fadeeva et al., 2023) | 0.149 | 0.203 | 0.819 | 0.104 | 0.186 | 0.774 | 0.243 | 0.253 | 0.764 | 0.252 | 0.268 | 0.790 |
| SC-VC (Xiong et al., 2024) | 0.299 | 0.325 | 0.704 | 0.451 | 0.474 | 0.559 | 0.122 | 0.211 | 0.761 | 0.362 | 0.378 | 0.653 |
| SC (Xiong et al., 2024) | 0.236 | 0.244 | 0.785 | 0.233 | 0.229 | 0.780 | 0.065 | 0.177 | 0.808 | 0.303 | 0.304 | 0.713 |
| NVC | 0.171 | 0.190 | 0.853 | **0.084** | **0.164** | 0.806 | 0.168 | 0.192 | 0.845 | 0.218 | 0.236 | 0.791 |
| DINCO | **0.097** | **0.155** | **0.879** | 0.177 | 0.179 | **0.835** | **0.044** | **0.148** | **0.864** | **0.121** | **0.191** | **0.817** |

Table 2: SimpleQA results. The black-box variants of our methods assume no logit access. Metrics and text styling follow Table 1.

| Method | GPT-4.1 | | | Gemini-2.5-Flash | | |
|---|---|---|---|---|---|---|
| | ECE↓ | BS↓ | AUC↑ | ECE↓ | BS↓ | AUC↑ |
| VC (Kadavath et al., 2022) | 0.547 | 0.549 | 0.644 | 0.409 | 0.393 | 0.617 |
| K-VC (Tian et al., 2023) | 0.338 | 0.337 | 0.632 | 0.535 | 0.511 | 0.566 |
| MSP (Fadeeva et al., 2023) | 0.263 | 0.255 | **0.800** | 0.098 | 0.177 | **0.773** |
| SC-VC (Xiong et al., 2024) | 0.223 | 0.252 | 0.761 | 0.186 | 0.221 | 0.755 |
| SC (Xiong et al., 2024) | 0.220 | 0.252 | 0.750 | 0.170 | 0.212 | 0.748 |
| NVC_black-box | 0.213 | 0.270 | 0.607 | 0.208 | 0.262 | 0.595 |
| DINCO_black-box | 0.161 | 0.251 | 0.605 | **0.079** | 0.199 | 0.697 |
| NVC | 0.164 | 0.222 | 0.729 | 0.105 | 0.199 | 0.662 |
| DINCO | **0.089** | **0.183** | 0.786 | 0.088 | **0.174** | 0.762 |

an ECE of 0.049 and 0.060 on TriviaQA and SimpleQA, respectively) despite only leveraging coherence in validation and not in generation (Section 2.3). Nonetheless, DINCO is more consistently calibrated than NVC (in particular on AUC, e.g. 0.786 DINCO vs. 0.729 NVC with GPT-4.1 on SimpleQA), empirically supporting our motivation in Section 2.3 for integrating coherence in generation (SC) and validation (NVC) into DINCO. In the black-box setting on SimpleQA, DINCO continues to do well (e.g. outperforming the baselines on ECE), but it tends to fall behind DINCO with logit access, underscoring the benefit of leveraging token probabilities for calibration. In Appendix D.6, we evaluate Qwen3-32B on BioASQ (Krithara et al., 2023), extending these findings to the biomedical domain where expert knowledge is required.

**Long-form QA.** Table 3 reports results on FactScore. While VC is extremely miscalibrated (e.g. ECE of 0.433 with Qwen3-8B), DINCO is able to leverage incoherence across related claims to normalize verbalized confidence and achieve strong calibration. Whether SC or NVC performs better varies by the model Qwen3-8B or Gemma-3-4B-IT, but DINCO continues to outperform SC (0.076 vs. 0.162 ECE with Qwen3-8B, and 0.172 vs. 0.197 ECE with Gemma-3-4B-IT). Furthermore, DINCO is the method most strongly correlated with passage-level FactScore (e.g. improving Pearson and Spearman correlation over SC by an average of 0.072 and 0.074, respectively), demonstrating that the effectiveness of DINCO extends to the long-form setting. Taken together with the short-form results in Tables 1 and 2, these results indicate that DINCO is applicable to open- and closed-source models, and crucially can transfer seamlessly between short-form QA and long-form generation settings.

Table 3: FactScore results. In addition to the claim-level metrics, we report Pearson ($r$) and Spearman ($\rho$) correlation with passage-level FactScore. Text styling follows Table 1, and we bold the best $r$ and $\rho$.

| Method | Qwen3-8B | | | | | Gemma-3-4B-IT | | | | |
|---|---|---|---|---|---|---|---|---|---|---|
| | ECE ↓ | BS ↓ | AUC ↑ | $r$ ↑ | $\rho$ ↑ | ECE ↓ | BS ↓ | AUC ↑ | $r$ ↑ | $\rho$ ↑ |
| VC (Kadavath et al., 2022) | 0.433 | 0.431 | 0.625 | 0.073 | 0.122 | 0.527 | 0.527 | 0.683 | -0.081 | -0.129 |
| SC (Zhang et al., 2024a) | 0.162 | 0.226 | **0.771** | 0.468 | 0.494 | 0.197 | 0.233 | 0.787 | 0.629 | 0.607 |
| NVC | 0.191 | 0.263 | 0.681 | 0.444 | 0.443 | **0.123** | 0.230 | 0.726 | 0.695 | 0.704 |
| DINCO | **0.076** | **0.202** | 0.767 | **0.518** | **0.538** | 0.172 | **0.210** | **0.793** | **0.724** | **0.712** |

Table 4: Saturation analysis (higher $\Delta$ = lower saturation). DINCO alleviates saturation.

| Method | $\Delta_0$ | $\Delta_{0.001}$ |
|---|---|---|
| VC | 0.670 | 0.605 |
| SC | 0.734 | 0.734 |
| SC@100 | 0.832 | 0.832 |
| DINCO | 0.998 | 0.984 |

Table 5: Ablation of NLI-based weighting shows it is crucial to performance.

| Method | ECE ↓ | BS ↓ | AUC ↑ |
|---|---|---|---|
| NVC | **0.171** | **0.190** | **0.853** |
| w/o NLI | 0.358 | 0.335 | 0.778 |
| DINCO | **0.097** | **0.155** | **0.879** |
| w/o NLI | 0.130 | 0.185 | 0.810 |

## 4 DISCUSSION AND ANALYSIS

We conduct further analysis using Qwen3-8B on TriviaQA with $P(\text{True})$, which appeared in Table 1 in the main experiments.

**Scaling Self-Consistency.** While our main experiments in Section 3 were approximately controlled for the inference budget of each method, here we examine whether simply scaling the inference budget of self-consistency allows it to recover the calibration of DINCO. Fig. 4 shows that self-consistency alone is unable to reach the performance of DINCO (using 5 distractors and 5 self-consistency samples as in Section 3) even when scaling up to 100 samples. Although DINCO uses 32% more FLOPs than SC in the context of Section 3, scaling SC up to match and far exceed this slightly higher FLOP count fails to further improve calibration, demonstrating that the effectiveness of DINCO comes from leveraging coherence in both generation and validation, which is not matched by scaling the generation axis alone.

In Appendix D.7, we show that combining distractor generation and validation into one step maintains similar calibration to the original two-step version while reducing cost to only 10% more than SC. In Appendix C.1, we motivate the feasibility of generating distractors using a smaller model to reduce cost below even that of SC.

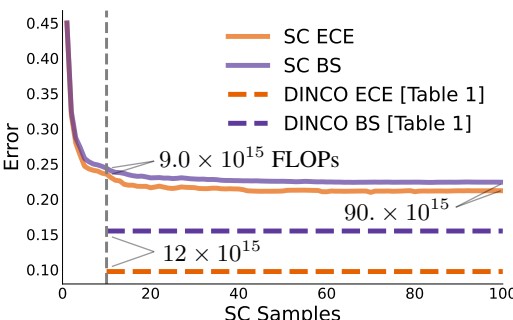

Figure 4: Scaling self-consistency does not close the gap with DINCO. FLOP counts are with Qwen3-8B on 1000 TriviaQA questions. Since each distractor requires a verbalized confidence step, DINCO costs 32% more than SC. However, due to diminishing returns, scaling SC up to even 100 samples negligibly improves performance (despite costing 7.6 times as much as DINCO), justifying the slightly higher cost of DINCO to achieve stronger calibration. We remark that the cost of the lightweight NLI model (184M parameters) used in DINCO and SC is negligible, making up less than 1% of the total FLOP count.

**Quantifying Saturation.** A core motivation for our method is the notion (shown in Fig. 1) that verbalized confidence exhibits saturation at high confidence. To better quantify this notion, we introduce a metric $\Delta_\epsilon$ that measures the absence of saturation. We define $\Delta_\epsilon$ as the proportion of pairs of distinct instances that have a confidence difference exceeding $\epsilon$. For example, if all confidence scores are the same, then $\Delta_0 = 0$, and if all confidence scores are distinct, then $\Delta_0 = 1$.

We consider $\epsilon \in \{0, 0.001\}$. Table 4 shows that DINCO leads to substantially higher rates of distinct confidence, indicating lower saturation. In particular, self-consistency scaled to 100 samples (as above) continues to be more saturated than DINCO. While the absence of saturation alone means little without calibration (as evaluated in Section 3), this analysis helps explain DINCO's calibration improvement, as hinted at by Fig. 1. Moreover, we argue that lower saturation leads to more usable confidence estimates: a saturated distribution is inherently less controllable, with large jumps in error between thresholds.

**Ablating NLI.**   Our main experiments in Tables 1 to 3 showed comparisons with SC and NVC, which ablate NVC and SC, respectively, from DINCO. Here, to understand how necessary access to an NLI model is, we ablate the NLI component, which is used to downweight distractors that overlap with the main claim or other distractors (Section 2.3, Fig. 2). In Table 5, we see that performance substantially decreases without NLI-based weighting, emphasizing the utility of an off-the-shelf NLI model to account for claim redundancy. In Appendix D.3, we show that DINCO is robust to the specific choice of the NLI model.

# 5   RELATED WORK

**Considering Multiple Answers for Verbalized Confidence.**   The approach most related to DINCO is to have the model consider several candidates within a single prompt and assign confidences to them (Tian et al., 2023; Kadavath et al., 2022; Zhang et al., 2024b; Chhikara, 2025). A subtle but crucial distinction between these methods and DINCO is that if we present all the candidates together, we become unable to gauge the probabilistic coherence of the confidence estimates, i.e. whether they form a valid probability distribution, since LLMs can satisfy probabilistic coherence via simple arithmetic. In Section 2.2, we show that the probabilistic coherence of confidence estimates is correlated with answer correctness. Instruction-tuned LLMs have a tendency to assert confidence even when it is undue (OpenAI, 2023; Leng et al., 2025; Sun et al., 2025; Xu et al., 2025), leading joint prompting to suffer similar issues of overconfidence as vanilla prompting. With independent prompting, we can expose and account for inconsistencies in self-declared knowledge. In Section 3, we empirically verify that our method leads to better calibration than joint prompting for verbalized confidence.

**Confidence Estimation in Long-form Generation.**   While historically confidence estimation has mostly been applied to classification (Houlsby et al., 2011), multiple-choice (Jiang et al., 2021), and short-form QA (Xiong et al., 2024), with the advent of LLMs it has increasingly been considered for long-form generation. Since token-level uncertainty is often ill-suited for representing claim-level uncertainty, the primary approaches have been self-consistency and verbalized confidence (Manakul et al., 2023; Zhang et al., 2024a; 2025). We propose a method to normalize verbalized confidence, enabling DINCO to combine the complementary confidence signals in these two prior approaches.

**Reconciling Inconsistent LLM Probability Judgments.**   In Appendix A.2, we discuss a related line of work demonstrating the benefits of reconciling inconsistent LLM probability judgments. In this work, we propose a zero-resource confidence estimator that normalizes verbalized confidence over self-generated distractors, motivated by the suggestibility of LLMs in unfamiliar topics.

# 6   CONCLUSION

We present DINCO, which estimates LLM confidence by leveraging coherence in generation as well as validation (through verbalized confidence). Verbalized confidence tends to be saturated at high confidence. We show evidence that this behavior is correlated with *suggestibility*, where the LLM is more likely to accept claims that it knows less about. Motivated by this finding, DINCO has the LLM verbalize its confidence independently on several self-generated distractors to estimate and correct for the bias caused by suggestibility. DINCO outperforms existing methods in the zero-resource setting on short-form QA (TriviaQA, SimpleQA) and long-form QA (FactScore) and mitigates saturation.

## ACKNOWLEDGMENTS

The authors acknowledge the Texas Advanced Computing Center (TACC) at The University of Texas at Austin for providing computational resources that have contributed to the research results reported within this paper.

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

# A  RELATED WORK

## A.1  RELATED BEHAVIOR IN LLMS AND HUMANS

Humans are also known to be susceptible to suggestibility. They can alter their memories to match the suggestions of other people, especially at a young age (Bruck & Ceci, 1999). Sycophancy is a similar behavior observed in LLMs, where an epistemically vacuous prompt such as *"Are you sure?"* often leads the model to change its answer (Laban et al., 2024). The user expressing doubt suggests to the model that its answer may be incorrect, since the user has some assumed level of credibility and would be unlikely to ask again if they agreed. Since instruction-tuned models aim to adhere to user preferences, it is plausible that they would employ an *argument from ignorance* (Walton, 1996) to accept a user claim that they cannot refute.

Zhu & Griffiths (2024) provide evidence of probabilistic incoherence in LLMs and attribute this finding to the prior from the Bayesian Sampler model, which has been used to explain incoherence in human probability judgments (Zhu et al., 2019). In particular, if the same prior is used for every probability judgment, the sum of the probability judgments for mutually exclusive events can exceed 1 (Zhu & Griffiths, 2024). This failure to satisfy the axioms of probability is consistent with our empirical evidence in Section 2.2.

## A.2  RECONCILING INCONSISTENT LLM PROBABILITY JUDGMENTS.

Prior work has demonstrated the benefits of reconciling inconsistent LLM probability judgments instead of taking them at face value. Jung et al. (2022) improve factuality by selecting claims to which the LLM assigns coherent truth values upon negation. Hou et al. (2025) use belief tree propagation with logically related claims to detect hallucinations. Nafar et al. (2025) find that independent prompting followed by normalization outperforms joint prompting for Bayesian network parameter estimation. Feng et al. (2025); Xia et al. (2024) optimize a probability distribution to approximately satisfy LLM-generated probability constraints. In this work, we propose a zero-resource confidence estimator that normalizes verbalized confidence over self-generated distractors, motivated by the suggestibility of LLMs in unfamiliar topics.

# B  EXPERIMENTAL SETUP

## B.1  EVALUATION METRICS

We adopt the notation from Section 2.1. For a claim $c$, the truth value is $v(c) \in \{0, 1\}$ and the assigned confidence is $f(c) \in [0, 1]$.

**Expected Calibration Error (ECE; Naeini et al., 2015).** The confidence space $[0, 1]$ is partitioned into $K$ intervals of equal length. Out of the $N$ claims in the dataset, let $B_k$ be the list of claims assigned a confidence in the interval $I_k = (\frac{k-1}{K}, \frac{k}{K}]$ (with $I_1$ including 0). We compute

bin-level truthfulness and confidence as

$$\bar{v}_k = \frac{1}{|B_k|} \sum_{c \in B_k} v(c), \qquad\qquad \bar{f}_k = \frac{1}{|B_k|} \sum_{c \in B_k} f(c). \tag{5}$$

ECE is the bin size-weighted average of the absolute differences:

$$\text{ECE} = \sum_{k=1}^{K} \frac{|B_k|}{N} |\bar{v}_k - \bar{f}_k| \tag{6}$$

**Brier score (BS; Brier, 1950).** For a dataset of claims $c_1, \ldots, c_N$, the Brier score is the mean squared error between the truth values and the confidence estimates:

$$\text{BS} = \frac{1}{N} \sum_{i=1}^{N} (v(c_i) - f(c_i))^2 \tag{7}$$

**Area under ROC curve (AUC; Hanley & McNeil, 1982)** The ROC curve (depicted in Fig. 1 right) captures the tradeoffs between true and false positive rate (TPR, FPR) that we can obtain with selective prediction, i.e. setting a confidence threshold above which to accept claims. We take correct and incorrect claims to be labeled positive and negative, respectively. The TPR is the proportion of positive instances that are accepted, and the FPR is the proportion of negative instances that are accepted. We want the TPR to be high but the FPR to be low. By setting a lower confidence threshold, TPR will be higher, but FPR may also be higher. By setting a higher confidence threshold, FPR will be lower, but TPR may also be lower. As not every TPR (or FPR) in the interval $[0, 1]$ may be achievable, the ROC fills in the gaps between achievable (TPR, FPR) tradeoffs with linear interpolations, as seen with verbalized confidence in Fig. 1. As a summary statistic for the selective predictive power that a confidence estimator grants us, we compute the area under the ROC curve (AUC). For example, if all positive instances are assigned a higher confidence than all negative instances, the AUC is 1. Meanwhile, if confidences are sampled independently at random from the same distribution, the expected AUC is 0.5.

The AUC can also be characterized as the probability that a random positive instance is assigned higher confidence than a random negative instance, with ties randomly broken. Denoting $C_+$ and $C_-$ as the list of correct and incorrect claims in the dataset, respectively,

$$\text{AUC} = \frac{1}{|C_+||C_-|} \sum_{c_+ \in C_+} \sum_{c_- \in C_-} \frac{\mathbf{1}\{f(c_+) \geq f(c_-)\} + \mathbf{1}\{f(c_+) > f(c_-)\}}{2}. \tag{8}$$

## B.2 PROMPTS

### B.2.1 SHORT-FORM QA

> Prompt to generate main answer. Also used for beam search in DINCO and sampling for self-consistency.
>
> Here are 2 sets of example prompt and answer.
>
> Example Prompt: Which American-born Sinclair won the Nobel Prize for Literature in
> ↪   1930?
> Example Answer: Sinclair Lewis
>
> Example Prompt: Where in England was Dame Judi Dench born?
> Example Answer: York
>
> ---
>
> Now, here is a new prompt to answer. Answer with a concise phrase, as in the examples.
>
> Prompt: {question}
> Answer:

---

**P(True)**

Below is a question and a candidate answer. Your task is to determine whether the answer is
↪    correct or not. Only output "Yes" (correct) or "No" (incorrect).

Question: {question}
Candidate answer: {candidate_answer}

---

**Verbalized numerical confidence**

Below is a question and a candidate answer. State your confidence that the candidate
↪    answer is correct. Only output an integer followed by "%".

Question: {question}
Candidate answer: {candidate_answer}

---

**$K$-VC (Verb. 1S from Tian et al., 2023). We use $K = 10$.**

Provide your {K} best guesses and the probability that each is correct (0.0 to 1.0) for the
↪    following question. Give ONLY the guesses and probabilities, no other words or
↪    explanation. For example:

G1: <first most likely guess, as short as possible; not a complete sentence, just the guess!>
P1: <the probability between 0.0 and 1.0 that G1 is correct, without any extra commentary
↪    whatsoever; just the probability!>
...
G{K}: <{K}th most likely guess, as short as possible; not a complete sentence, just the
↪    guess!>
P{K}: <the probability between 0.0 and 1.0 that G{K} is correct, without any extra
↪    commentary whatsoever; just the probability!>

The question is: {question}

---

**Follow-up after main answer for SC-VC**

Is your answer correct? Only output "Yes" or "No".

---

### B.2.2    BIOGRAPHY GENERATION

---

**Prompt to generate main biography. Also used to sample biographies for self-consistency.**

Write me a paragraph biography on {entity}.

---

---

**Prompt to generate one distractor. We use beam search to extract multiple.**

You will be given a fact about a person. Assuming the fact is accurate, your task is to
↪    generate a plausible but inaccurate statement of a similar nature. The distractor
↪    statement should form a minimal pair with the original statement, i.e. the distractor
↪    should be as similar to the original as possible while ensuring that the distractor is not
↪    factual. The distractor should be crafted so that someone with only superficial
↪    knowledge about the topic is likely to be fooled.

Let's see some examples before the real task.

Topic: Barack Obama
Fact: Barack Obama was born in Hawaii.
Distractor: Barack Obama was born in Kenya.

Topic: Wright brothers
Fact: Wright airplanes were involved in fatal crashes.
Distractor: Wright airplanes were praised for their safety.

Topic: John Clempert
Fact: John Clempert was inspired by Houdini when developing acts.
Distractor: John Clempert was inspired by Penn and Teller when developing acts.

Now for the real task. Only output a distractor as in the examples.

Topic: {entity}
Fact: {claim}
Distractor:

---

**P(True)**

Your task is to determine whether the following claim related to {entity} is correct.
↪    Only output "Yes" (correct) or "No" (incorrect).

Claim: {claim}

Yes or No:

---

**Verbalized numerical confidence**

The claim below was found in a passage about {entity}. State your confidence that the
↪    claim is correct. Only output an integer followed by "%".

Claim: {claim}

---

---

**Prompt for the LLM to determine whether a sampled biography entails a claim for self-consistency**

---

You will be given a passage and a claim. Your task is to determine whether the passage
↪    supports, refutes, or does not mention the claim. Output only "Support", "Refute", or
↪    "No Mention".

Let's see some examples before the real task.

Passage: Barack Obama was the 44th President of the United States, serving from 2009 to
↪    2017. Born on August 4, 1961, in Honolulu, Hawaii, he was the first African
↪    American to hold the office. Before his presidency, Obama served as a state senator in
↪    Illinois and later as the 47th Governor of Illinois. A former constitutional law
↪    professor, he was known for his eloquence, bipartisan approach, and focus on issues
↪    such as healthcare reform, climate change, and foreign policy. His presidency was
↪    marked by significant legislative achievements, including the Affordable Care Act,
↪    and a commitment to diplomacy and international cooperation. After leaving office, he
↪    authored memoirs and remained active in public life, advocating for social justice and
↪    community engagement.
Claim: Barack Obama was born in Hawaii.
Relationship: Support

Passage: Tiger Woods is one of the most iconic and accomplished golfers in history, known
↪    for his extraordinary talent, dominance on the course, and global influence on the
↪    sport. Born on December 30, 1975, in Cypress, Florida, Woods rose to fame in the
↪    mid-1990s and quickly became a household name, winning his first major
↪    championship at the 1997 Masters at just 21 years old. Over his career, he has claimed
↪    15 major titles, the most in PGA Tour history, and has consistently ranked among the
↪    world's top golfers for over two decades. His aggressive playing style, precision, and
↪    mental toughness set him apart, making him a symbol of excellence in golf. Despite
↪    personal challenges and setbacks, Woods has remained a dominant force in the sport,
↪    inspiring millions of fans around the world.
Claim: Tiger Woods won a major championship at 19 years old.
Relationship: Refute

Passage: Albert Einstein was a theoretical physicist renowned for developing the theory of
↪    relativity, which revolutionized the understanding of space, time, and gravity. Born in
↪    1879 in Ulm, Germany, he later moved to Switzerland and eventually to the United
↪    States. Einstein's work, including the famous equation E=mc², laid the foundation for
↪    modern physics and contributed to the development of nuclear energy. Despite his
↪    scientific achievements, he was also a passionate advocate for peace, civil rights, and
↪    education. His legacy endures as one of the most influential scientists in history.
Claim: Albert Einstein became a US citizen.
Relationship: No Mention

Now for the real task.

Passage: {sampled_biography}
Claim: {claim}
Relationship:

---

### B.2.3 PSEUDO-BEAM SEARCH

> **Prompt for the LLM to complete the prefix of an answer. Used for pseudo-beam search (Appendix C.2).**
>
> You will be given a prompt along with a prefix to begin your answer with. Your answer
> ↪   should start with the given prefix. If the prefix itself is your final answer, you can
> ↪   simply output just the prefix.
>
> Let's look at 2 examples before the real task.
>
> Example Prompt: Which American-born Sinclair won the Nobel Prize for Literature in
> ↪   1930?
> Example Answer Prefix: Sin
> Example Answer: Sinclair Lewis
>
> Example Prompt: Where in England was Dame Judi Dench born?
> Example Answer Prefix: York
> Example Answer: York
>
>
> ---
>
> Now, here is a new prompt to answer. Answer with a concise phrase starting with the given
> ↪   prefix, as in the examples.
>
> Prompt: {question}
> Prefix: {prefix}
> Answer:

### B.3 LLM-AS-A-JUDGE

We score model responses for short-form QA (TriviaQA and SimpleQA) using an LLM judge rather than lexical matching for robust evaluation (Wei et al., 2024). Our biography generation task is evaluated with FactScore (Min et al., 2023), which uses a strong LLM for atomic claim decomposition and verification; we use GPT-4.1. We use Llama-3.1-8B-Instruct (Dubey et al., 2024) as the judge on TriviaQA, and GPT-4.1 as the judge on SimpleQA. On a sample of 100 questions from each dataset, we compared LLM judgments with human judgments from one author of this paper. We used responses from Qwen3-8B on TriviaQA, and responses from GPT-4.1 on SimpleQA. The rate of agreement was 96/100 and 99/100, respectively. Upon reviewing disagreements, we found that in most cases there was genuine room for interpretation. For example, for the question *"In The Living Daylights what did Carla keep in her cello case?"*, the target answer is *"A machine gun"* while the model answer was *"A gun"*. It is unclear whether the model answer has the desired specificity. As another example, for the question *"In Charles Dickens' "Great Expectations", who or what was Abel Magwitch?"*, the target answer is *"Convict"* while the model answer was *"A convict and the main benefactor of Pip"*. The model answer contains the answer but contains more information, and without external information, it is impossible to determine whether the model's answer is correct. Overall, given the high agreement on unambiguously gradable questions, we deem it safe to adopt LLM-as-a-judge as a reliable evaluator in our experiments.

### B.4 FACTSCORE EXAMPLE

Table 6 presents an example of atomic claim decomposition and verification with FactScore.

### B.5 SIGNIFICANCE TESTING

**ECE.** We subsample 10k subsets of size 0.9 times the original dataset, where sampling is done without replacement. We construct an upper one-sided confidence interval with confidence level

Table 6: Example of FactScore atomic claim decomposition and verification.

**Generation**

Kang Ji-hwan is a renowned South Korean actor and singer, best known for his role as the lead vocalist of the popular K-pop group BE:FIRST. Born on April 15, 2001, in Seoul, South Korea, Kang began his career in the entertainment industry at a young age, showcasing his talent through various music projects and performances. His distinctive voice and charismatic stage presence quickly earned him a loyal fanbase. Beyond his work in music, Kang has also ventured into acting, appearing in television dramas and variety shows, further solidifying his status as a multifaceted entertainer. With his dedication and natural talent, Kang Ji-hwan continues to make a significant impact in the K-pop and entertainment world.

| Extracted claim | Correct? |
| --- | --- |
| Kang Ji-hwan is a South Korean actor. | Yes |
| Kang Ji-hwan is a South Korean singer. | No |
| Kang Ji-hwan is renowned. | Yes |
| Kang Ji-hwan is best known for his role as the lead vocalist of BE:FIRST. | No |
| BE:FIRST is a K-pop group. | Yes |
| BE:FIRST is a popular group. | Yes |
| Kang was born on April 15, 2001. | No |
| Kang was born in Seoul, South Korea. | Yes |
| Kang began his career in the entertainment industry at a young age. | No |
| Kang has showcased his talent through various music projects. | No |
| Kang has showcased his talent through various performances. | Yes |
| He has a distinctive voice. | No |
| He has a charismatic stage presence. | No |
| His distinctive voice quickly earned him a loyal fanbase. | No |
| His charismatic stage presence quickly earned him a loyal fanbase. | No |
| He quickly earned a loyal fanbase. | Yes |
| Kang has worked in music. | No |
| Kang has ventured into acting. | Yes |
| Kang has appeared in television dramas. | Yes |
| Kang has appeared in variety shows. | Yes |
| Kang is a multifaceted entertainer. | Yes |
| Kang's status as a multifaceted entertainer has been further solidified. | Yes |
| Kang Ji-hwan is dedicated. | Yes |
| Kang Ji-hwan has natural talent. | No |
| Kang Ji-hwan continues to make a significant impact in the K-pop world. | No |
| Kang Ji-hwan continues to make a significant impact in the entertainment world. | No |

0.95 for the tested method's ECE minus the best method's ECE and check whether the interval contains 0.

**BS.** As the Brier score is simply the mean squared error between confidences and truth values, it is well behaved and amenable to bootstrapping. We sample 10k subsets with the same size as the original dataset, where sampling is done with replacement. We construct an upper one-sided confidence interval with confidence level 0.95 for the tested method's BS minus the best method's BS and check whether the interval contains 0.

**AUC.** As AUC is a U-statistic, we use a one-sided DeLong test (DeLong et al., 1988; Sun & Xu, 2014) with confidence level 0.95.

## C Methods

### C.1 Examples, Analysis, and Discussion of Distractors

**Examples.** Building on the example in Fig. 2, we provide more examples of distractors generated by LLMs using different methods of self-generating distractors: beam search on a vanilla QA prompt (Table 7), beam search on a prompt eliciting one distractor for a given claim (Table 8), pseudo-beam search on a vanilla QA prompt (Table 9a), and a prompt eliciting a list of candidate answers (Table 9b); see Sections 2.3 and 3 for further descriptions and Appendix B.2 for prompts. For each distractor, we also report the verbalized confidence, $w_{\text{unique}}$, and $w_{\text{contra}}$, which underlie the computation of $\beta(C)$ in Eq. 4 and measure plausibility, uniqueness, and counterfactuality, respectively. We report these quantities under the experimental settings in Section 3.

**Quantitative Analysis.** We analyze these attributes of self-generated distractors over a full dataset. Fig. 5 shows that beam search on a vanilla QA prompt leads to distractors with high mean values of $P(\text{True})$ (indicating plausibility), $w_{\text{unique}}$ (indicating low repetitiveness), and $w_{\text{contra}}$ (indicating opposition to the main claim). In other words, beam search on a vanilla QA prompt enables efficient and diverse exploration of alternative claims, despite not explicitly prompting for distractors. We summarize this analysis for the other distractor generation methods in Fig. 6, where we consistently observe high average values of verbalized confidence, $w_{\text{unique}}$, and $w_{\text{contra}}$, suggesting that many self-generated distractors fulfill the desiderata of being plausible, unique, and counterfactual.

**Discussion.** Identifying entailment and contradiction relationships may be a nontrivial task in general, such as in cases requiring domain expertise. DINCO does not dictate the choice of the NLI model, so in such cases, it may be worth using a stronger NLI model or even the generator LLM itself. While this reliance on the NLI capabilities of the same LLM we are attempting to calibrate may seem circular, we argue that NLI is a tractable and modular task that is largely solved, in contrast to LLM calibration. In this paper, we empirically show that a lightweight off-the-shelf NLI model suffices to achieve strong calibration across short-form and long-form generation settings. In Appendix D.6, we show that the same lightweight NLI model continues to suffice in the biomedical domain despite its highly technical nature. In Appendix D.3, we show that DINCO is robust to the choice of the NLI model.

DINCO uses the generator LLM to generate its own distractors to relieve the need for external models. However, one concern is that a model's epistemic uncertainty may limit its ability to generate plausible, high-quality distractors. Fortunately, our quantitative analysis (Figs. 5 and 6) provides empirical evidence of the plausibility of generated distractors. To offer an intuitive explanation of this behavior, we note that the "plausibility" of a claim depends on the model assessing this plausibility. If a model is epistemically uncertain, the distractors it generates may be implausible to an expert model that is epistemically *certain*, but this is acceptable because we are interested in whether the distractors are plausible to the epistemically *uncertain* model so that we can leverage the phenomenon of suggestibility under epistemic uncertainty (Section 2.1). As an extreme case of this, it is even acceptable for a distractor to be factual, as long as it is counterfactual to the main claim. As shown by the examples in Table 9b, we want to know whether the model can distinguish correct and incorrect claims, regardless of whether the distractors happen to be correct.

Table 7: Example distractors and their attributes. Qwen3-8B on TriviaQA. Main answer and distractors generated using beam search on a vanilla QA prompt (Appendix B.2.1).

| Question | | | |
|---|---|---|---|
| In The Living Daylights what did Carla keep in her cello case | | | |

| True answer | | | |
|---|---|---|---|
| A machine gun | | | |

| Main answer | VC | $\beta(C)$ | NVC |
|---|---|---|---|
| A gun | 1.00 | 5.91 | 0.17 |

| Distractor | VC | $w_{\text{unique}}$ | $w_{\text{contra}}$ |
|---|---|---|---|
| A bomb | 1.00 | 1.00 | 1.00 |
| A jar of honey | 0.12 | 1.00 | 1.00 |
| A set of keys | 0.92 | 1.00 | 1.00 |
| A jar of spiders | 0.82 | 0.99 | 1.00 |
| A stash of drugs | 1.00 | 1.00 | 1.00 |
| A pair of binoculars | 0.01 | 1.00 | 1.00 |
| A jar of pickles | 0.99 | 0.50 | 1.00 |
| A jar of pickled gherkins | 1.00 | 0.51 | 1.00 |
| A secret weapon | 0.97 | 0.54 | 0.10 |

Table 8: Example distractors and their attributes. Qwen3-8B on FactScore (biography generation). The main claim is extracted from the generated biography, and the distractors are generated using beam search on a prompt eliciting one distractor given the main claim (Appendix B.2.2).

| Main claim (entity: Kalki Koechlin) | VC | $\beta(C)$ | NVC |
|---|---|---|---|
| She played Nandini in the film Chandni Chowk to China. | 0.49 | 1.48 | 0.33 |

| Distractor | VC | $w_{\text{unique}}$ | $w_{\text{contra}}$ |
|---|---|---|---|
| She played Chandni in the film Chandni Chowk to China. | 0.07 | 0.50 | 0.98 |
| She played Nandini in the film Chandani Chowk to China. | 0.81 | 0.98 | 0.00 |
| She played Nandini in the film Chandni Chowk to Russia. | 0.45 | 0.99 | 1.00 |
| She played Nandini in the film Titanic. | 0.00 | 1.00 | 1.00 |
| **** She played Chandni in the film *Chandni Chowk to China*. | 1.00 | 0.50 | 0.98 |
| She played Nandini in the film Chandni Chowk to Italy. | 0.01 | 0.99 | 1.00 |
| She played Nandini in the film Dangal. | 0.00 | 1.00 | 0.99 |
| She played Nandini in the film Chandni Chowk to Canada. | 0.00 | 0.99 | 1.00 |

The discussion above for the well-suitedness of a model to generate its own distractors also hints at the potential for distractors to be generated by a smaller model to reduce computational cost. Although we do not investigate this idea in this paper, we expect the task of generating somewhat plausible distractors to be much easier than knowing the correct answer. Considering the example in Table 9, the information required to correctly answer the question is obscure, but anyone who understands the question can produce distractors that would likely be indistinguishable from the correct answer to someone else who does not know the answer to the question. This suggests that a model with a smaller capacity than the model we seek to calibrate could be used for efficient distractor generation with little degradation in distractor quality. Outside of calibration, other methods such as contrastive decoding (Li et al., 2023) leverage a similar intuition that weaker models can produce plausible but incorrect candidate generations.

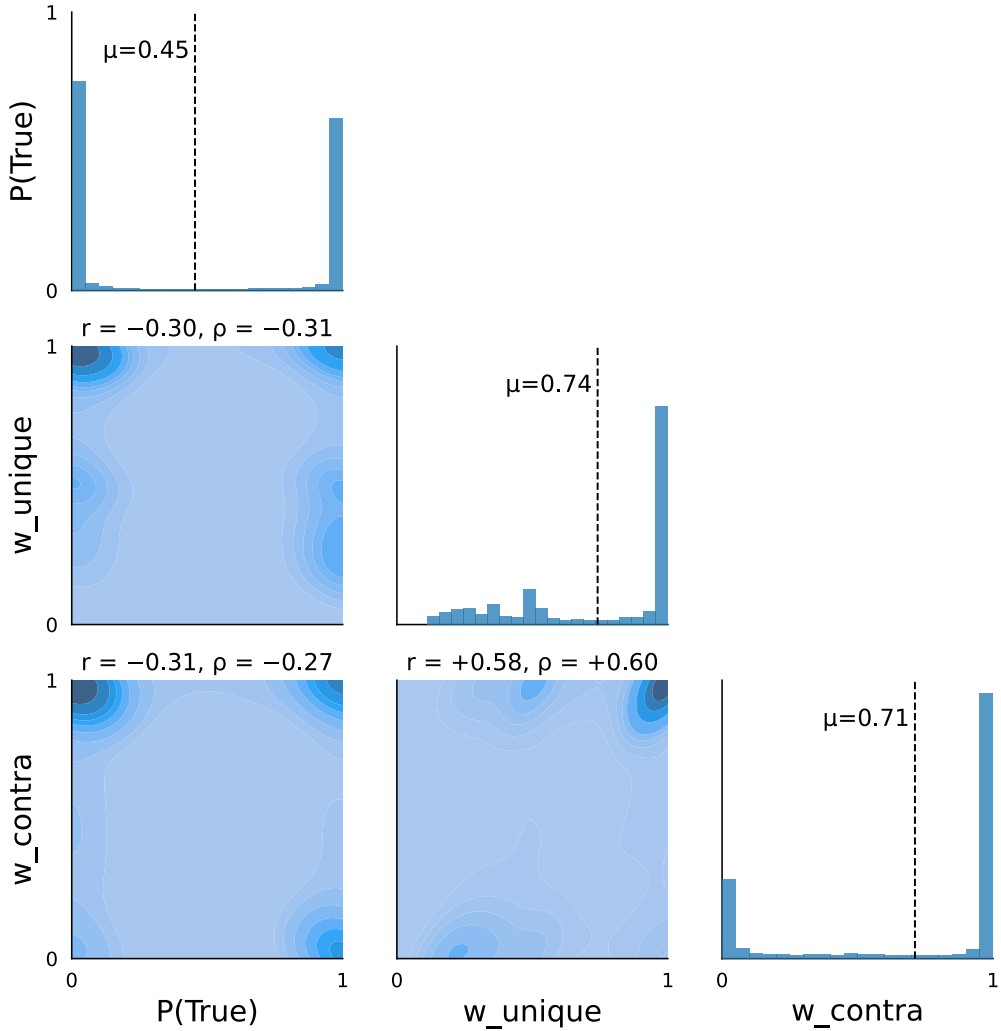

Figure 5: Pairwise correlations (Pearson's $r$, Spearman's $\rho$) between distractor attributes: $P(\text{True})$, $w_{\text{unique}}$, and $w_{\text{contra}}$. Due to the large numbers of points, we depict joint distributions by plotting the density using kernel density estimation, with darker shades indicating more points. The diagonals plot the marginal distributions. These results are with Qwen3-8B on TriviaQA, using beam search on a vanilla QA prompt (Appendix B.2.1) to generate distractors. In other words, we do not explicitly prompt the LLM to generate incorrect answers, motivating our investigation of the plausibility, uniqueness, and counterfactuality of the distractors. We find that $w_{\text{unique}}$ and $w_{\text{contra}}$ are positively correlated, indicating that distractors that contradict the main claim tend to be repeated fewer times within the distractor set, suggesting a diverse exploration of the set of alternative claims. On the other hand, $w_{\text{unique}}$ and $w_{\text{contra}}$ are negatively correlated with $P(\text{True})$. Interpreting $P(\text{True})$ as a measure of plausibility, this means that plausible claims are more repeatedly generated and are more in agreement with the main claim. Nevertheless, these negative correlations are weak, and the fact that the mean values of $P(\text{True})$ and $w_{\text{contra}}$ are both high indicates that a substantial number of plausible distractors are being generated.

Table 9: Example distractors and their attributes. GPT-4.1 on SimpleQA. Table 9a shows examples using pseudo-beam search on a vanilla QA prompt to generate distractors. Table 9b shows examples in the black-box setting using a single generation to a prompt asking for a list of candidate answers, and the verbalized confidence is verbalized numerical confidence rather than $P(\mathrm{True})$ (we still elicit confidence for each candidate answer independently). See Appendix B.2.1 for prompts.

| Question |
|---|
| For how many years did Mohamed Abdelaziz Djaït serve as the Mufti of the Republic of Tunisia? |

| True answer |
|---|
| 3 |

(a) Pseudo-beam search.

| Main answer | VC | $\beta(C)$ | NVC |
|---|---|---|---|
| 17 years | 0.85 | 6.19 | 0.14 |

| Distractor | VC | $w_{\mathrm{unique}}$ | $w_{\mathrm{contra}}$ |
|---|---|---|---|
| Twenty years | 0.78 | 0.52 | 0.99 |
| Two years | 0.62 | 1.00 | 1.00 |
| 12 years | 0.22 | 1.00 | 1.00 |
| 22 years | 0.12 | 0.54 | 1.00 |
| Three years | 0.90 | 1.00 | 1.00 |
| 11 years | 0.90 | 1.00 | 1.00 |
| Six years | 0.73 | 1.00 | 1.00 |
| Eight years | 0.78 | 1.00 | 1.00 |
| Four years | 0.73 | 1.00 | 1.00 |

(b) Black-box prompting.

| Main answer | VC | $\beta(C)$ | NVC |
|---|---|---|---|
| 2 | 1.00 | 9.56 | 0.10 |

| Distractor | VC | $w_{\mathrm{unique}}$ | $w_{\mathrm{contra}}$ |
|---|---|---|---|
| 3 | 1.00 | 1.00 | 0.99 |
| 1 | 1.00 | 1.00 | 0.99 |
| 4 | 1.00 | 1.00 | 1.00 |
| 5 | 0.80 | 1.00 | 1.00 |
| 6 | 1.00 | 1.00 | 1.00 |
| 7 | 1.00 | 1.00 | 1.00 |
| 8 | 1.00 | 1.00 | 1.00 |
| 9 | 1.00 | 1.00 | 1.00 |
| 10 | 0.80 | 1.00 | 1.00 |

## C.2 PSEUDO-BEAM SEARCH

DINCO uses beam search to generate distinct distractors. However, beam search may not be available for API-access models. Here, we describe an approximation of beam search with similar inference cost that can be implemented if top token probabilities are provided, as they are in many API-access models such as GPT and Gemini.

We first run one inference pass to generate the main answer, which also gives us the highest probability tokens (and their probabilities) at each token position in the main answer. We consider the set of sequences that are a prefix of the main answer followed by a top token that is not the token in the main answer. We sort the sequences by their probabilities (computed with the chain rule). For each of the highest probability sequences, we present it to the LLM as a prefix to complete as an answer to the question (see Appendix B.2.3 for the prompt). This procedure chooses distinct prefixes with relatively high probability, shaping their completions to be distinct as well as plausible answers.

## C.3 SELF-CONSISTENCY

Self-consistency estimates confidence in a main answer by sampling many answers and counting the proportion of answers that match the main answer (Xiong et al., 2024; Wang et al., 2023). Given a short-form question, a main answer $c_0$, and sampled answers $c_1, \ldots, c_K$, we compute the confidence of $c_0$ as

$$f^{\mathrm{SC}}(c_0) = \frac{1}{K+1} \sum_{k=0}^{K} \mathbf{1}\{c_0 = c_k\}. \tag{9}$$

Following Kuhn et al. (2023), we use an NLI model to determine semantic equivalence.[5] For the long-form setting, we follow Zhang et al. (2024a) and decompose a long-form response $r_0$ into claims $c_1, \ldots, c_n$ and measure entailment from other sampled responses $r_1, \ldots, r_K$. We compute the confidence $f^{\mathrm{SC}}(c_i)$ of a claim $c_i$ by Eq. 9 with the summand replaced with $P(\mathrm{entail} \mid r_k, c_i)$.

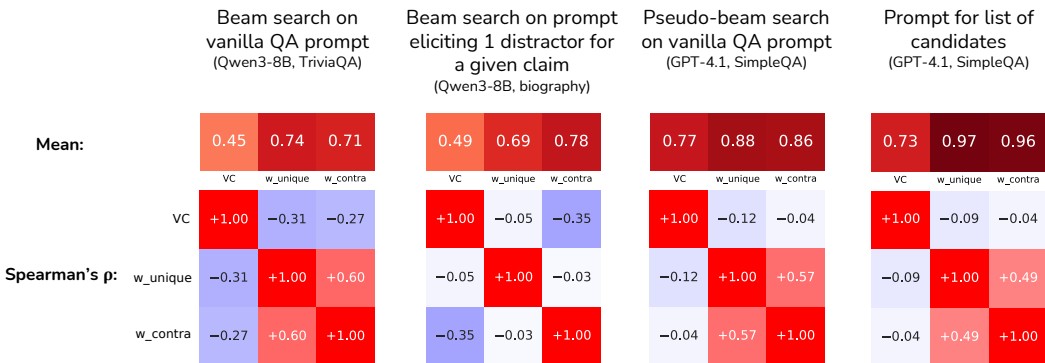

Figure 6: Statistics of distractor attributes (verbalized confidence, $w_{\mathrm{unique}}$, and $w_{\mathrm{contra}}$) for different methods of self-generating distractors. We report their means and Spearman correlations. Although the second method (beam search on a prompt eliciting one distractor for a given claim) has no correlation between $w_{\mathrm{unique}}$ and $w_{\mathrm{contra}}$ unlike the other methods, we find that all methods have high average values of verbalized confidence, $w_{\mathrm{unique}}$, and $w_{\mathrm{contra}}$, suggesting that many self-generated distractors fulfill the desiderata of being plausible, unique, and counterfactual. Notably, the fourth method (prompting for a list of candidate answers) has the highest $w_{\mathrm{unique}}$ and $w_{\mathrm{contra}}$, showing the potential benefit of conditioning on previously generated distractors rather than generating them independently. We note that the fourth method uses verbalized numerical confidence rather than $\mathrm{P(True)}$ due to being in the black-box setting (we still elicit confidence for each candidate answer independently).

Since the NLI task here involves a long-form text, we use the original LLM instead of an NLI model (see Appendix B.2 for prompts).

# D  ABLATIONS AND RESULTS

## D.1  PRELIMINARY STUDY EXTENDED

To extend the preliminary study presented in Section 2.2 comparing confidence levels on correctly and incorrectly answered questions, we provide the same analysis for the other two datasets in our main experiments (Section 3). SimpleQA (Wei et al., 2024) is a short-answer QA dataset similar to TriviaQA except designed to challenge frontier models. We accordingly evaluate GPT-4.1 on SimpleQA, thus extending our result to the scale of frontier models. Biography generation is evaluated with FactScore (Min et al., 2023). Similar to Fig. 3, Fig. 7 finds higher total confidence (over the main claim and distractors) on incorrectly answered questions, consistent with our hypothesis that LLMs are more susceptible to suggestibility when epistemically uncertain.

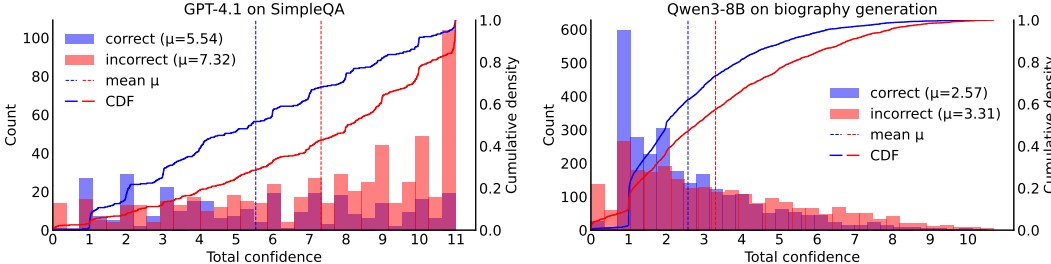

Figure 7: Distributions of total confidence over correct and incorrect answers, showing the same trends as Fig. 3 of higher total confidence on incorrectly answered questions. For each question, we generate a main answer and 10 distractors, so the maximum possible total confidence is 11.

Table 10: TriviaQA results. Building on Table 1, we report results when replacing P(True) with verbalized numerical confidence. Text styling follows Table 1.

| | Qwen3-8B | | | Qwen3-1.7B | | | Llama-3.2-3B-Instruct | | | Gemma-3-4B-IT | | |
|---|---|---|---|---|---|---|---|---|---|---|---|---|
| Method | ECE ↓ | BS ↓ | AUC ↑ | ECE ↓ | BS ↓ | AUC ↑ | ECE ↓ | BS ↓ | AUC ↑ | ECE ↓ | BS ↓ | AUC ↑ |
| $VC_{numerical}$ (Tian et al., 2023) | 0.310 | 0.304 | 0.791 | 0.364 | 0.342 | 0.697 | 0.226 | 0.226 | 0.773 | 0.311 | 0.321 | 0.731 |
| $VC_{P(True)}$ (Kadavath et al., 2022) | 0.240 | 0.242 | 0.817 | 0.387 | 0.383 | 0.720 | 0.189 | 0.208 | 0.826 | 0.300 | 0.299 | 0.702 |
| $K$-VC (Tian et al., 2023) | 0.341 | 0.348 | 0.604 | 0.538 | 0.524 | 0.596 | 0.146 | 0.228 | 0.678 | 0.254 | 0.262 | 0.786 |
| MSP (Fadeeva et al., 2023) | 0.149 | 0.203 | 0.819 | 0.104 | 0.186 | 0.774 | 0.243 | 0.253 | 0.764 | 0.252 | 0.268 | 0.790 |
| SC-VC (Xiong et al., 2024) | 0.299 | 0.325 | 0.704 | 0.451 | 0.474 | 0.559 | 0.122 | 0.211 | 0.761 | 0.362 | 0.378 | 0.653 |
| SC (Xiong et al., 2024) | 0.236 | 0.244 | 0.785 | 0.233 | 0.229 | 0.780 | 0.065 | 0.177 | 0.808 | 0.303 | 0.304 | 0.713 |
| $NVC_{numerical}$ | 0.232 | 0.225 | 0.870 | **0.069** | **0.162** | 0.812 | 0.183 | 0.202 | 0.825 | 0.267 | 0.272 | 0.787 |
| $DINCO_{numerical}$ | 0.117 | 0.171 | 0.857 | 0.170 | 0.183 | 0.825 | **0.037** | 0.150 | 0.857 | **0.101** | 0.202 | 0.791 |
| $NVC_{P(True)}$ | 0.171 | 0.190 | 0.853 | 0.084 | 0.164 | 0.806 | 0.168 | 0.192 | 0.845 | 0.218 | 0.236 | 0.791 |
| $DINCO_{P(True)}$ | **0.097** | **0.155** | **0.879** | 0.177 | 0.179 | **0.835** | 0.044 | **0.148** | **0.864** | 0.121 | **0.191** | **0.817** |

Table 11: FactScore results. Building on Table 3, we report results when replacing P(True) with verbalized numerical confidence. Text styling follows Table 3.

| | Qwen3-8B | | | | | Gemma-3-4B-IT | | | | |
|---|---|---|---|---|---|---|---|---|---|---|
| Method | ECE ↓ | BS ↓ | AUC ↑ | $r$ ↑ | $\rho$ ↑ | ECE ↓ | BS ↓ | AUC ↑ | $r$ ↑ | $\rho$ ↑ |
| $VC_{numerical}$ (Tian et al., 2023) | 0.327 | 0.328 | 0.749 | 0.508 | 0.565 | 0.482 | 0.465 | 0.648 | 0.186 | 0.169 |
| $VC_{P(True)}$ (Kadavath et al., 2022) | 0.433 | 0.431 | 0.625 | 0.073 | 0.122 | 0.527 | 0.527 | 0.683 | -0.081 | -0.129 |
| SC (Zhang et al., 2024a) | 0.162 | 0.226 | 0.771 | 0.468 | 0.494 | 0.197 | 0.233 | 0.787 | 0.629 | 0.607 |
| $NVC_{numerical}$ | 0.181 | 0.259 | 0.693 | 0.555 | 0.566 | **0.090** | 0.208 | 0.744 | 0.683 | 0.698 |
| $DINCO_{numerical}$ | **0.045** | **0.193** | **0.779** | **0.559** | **0.574** | 0.120 | **0.188** | **0.808** | 0.713 | 0.696 |
| $NVC_{P(True)}$ | 0.191 | 0.263 | 0.681 | 0.444 | 0.443 | 0.123 | 0.230 | 0.726 | 0.695 | 0.704 |
| $DINCO_{P(True)}$ | 0.076 | 0.202 | 0.767 | 0.518 | 0.538 | 0.172 | 0.210 | 0.793 | **0.724** | **0.712** |

## D.2 P(True) vs. Verbalized Numerical Confidence

In Tables 10 and 11, we expand on Tables 1 and 3 and report results on TriviaQA and biography generation when replacing P(True) with verbalized numerical confidence (see Appendix B.2 for prompts). For SimpleQA, the black-box variants of our methods in Table 2 use verbalized numerical confidence rather than P(True).

We find that the calibration benefits of DINCO generalize to the format of verbalized numerical confidence. For example, $DINCO_{numerical}$ compared to $VC_{numerical}$ lowers ECE by 0.196 on average on TriviaQA. $DINCO_{numerical}$ outperforms the stronger baselines as well, e.g. a lower ECE than SC by 0.103 on average on TriviaQA.

In the long-form setting of biography generation (Table 11), $DINCO_{numerical}$ continues to achieve strong calibration, even outperforming $DINCO_{P(True)}$ in many cases, e.g. 0.045 vs. 0.076 ECE with Qwen3-8B.

## D.3 NLI Model Choice

Our main experiments in Section 3 use the NLI model *DeBERTa-v3-base-mnli-fever-anli* (He et al., 2021). Here we ablate the particular choice of the NLI model. We choose NLI models that are lightweight (0.2-0.4B parameters) and frequently downloaded from HuggingFace. Table 12 confirms that DINCO and SC are robust to the choice of the NLI model, with DINCO consistently performing the best.

## D.4 Budget Hyperparameters

For our main experiments in Section 3, we gave each method an inference budget of $K = 10$, with DINCO splitting the budget evenly into 5 distractors and 5 SC samples. The equal budget split is a straightforward choice requiring no hyperparameter tuning and achieves strong calibration in Section 3. Here we examine how robust DINCO is to the budget split and whether tuning it

Table 12: NLI model choice ablation, building on Table 1 which uses the first NLI model. Text styling follows Table 1. DINCO and SC are robust to the choice of the NLI model, with DINCO consistently performing the best.

| Method | Qwen3-8B | | | Qwen3-1.7B | | | Llama-3.2-3B-Instruct | | | Gemma-3-4B-IT | | |
|---|---|---|---|---|---|---|---|---|---|---|---|---|
| | ECE↓ | BS↓ | AUC↑ | ECE↓ | BS↓ | AUC↑ | ECE↓ | BS↓ | AUC↑ | ECE↓ | BS↓ | AUC↑ |
| *NLI model:* `MoritzLaurer/DeBERTa-v3-base-mnli-fever-anli` *(184M parameters)* | | | | | | | | | | | | |
| SC (Xiong et al., 2024) | 0.236 | 0.244 | 0.785 | 0.233 | 0.229 | 0.780 | 0.065 | 0.177 | 0.808 | 0.303 | 0.304 | 0.713 |
| NVC | 0.171 | 0.190 | 0.853 | **0.084** | **0.164** | 0.806 | 0.168 | 0.192 | 0.845 | 0.218 | 0.236 | 0.791 |
| DINCO | **0.097** | **0.155** | **0.879** | 0.177 | 0.179 | **0.835** | **0.044** | **0.148** | **0.864** | **0.121** | **0.191** | **0.817** |
| *NLI model:* `sileod/deberta-v3-base-tasksource-nli` *(184M parameters)* | | | | | | | | | | | | |
| SC (Xiong et al., 2024) | 0.220 | 0.241 | 0.778 | 0.216 | 0.224 | 0.771 | 0.087 | 0.195 | 0.774 | 0.292 | 0.300 | 0.709 |
| NVC | 0.114 | 0.160 | 0.871 | **0.084** | **0.163** | 0.806 | 0.155 | 0.188 | 0.841 | 0.157 | 0.203 | 0.813 |
| DINCO | **0.108** | **0.157** | **0.879** | 0.189 | 0.184 | **0.835** | **0.052** | **0.148** | **0.864** | **0.137** | **0.193** | **0.821** |
| *NLI model:* `MoritzLaurer/mDeBERTa-v3-base-xnli-multilingual-nli-2mil7` *(279M parameters)* | | | | | | | | | | | | |
| SC (Xiong et al., 2024) | 0.225 | 0.245 | 0.771 | 0.212 | 0.226 | 0.765 | 0.098 | 0.204 | 0.757 | 0.289 | 0.298 | 0.712 |
| NVC | 0.165 | 0.190 | 0.855 | **0.081** | **0.173** | 0.788 | 0.192 | 0.203 | 0.836 | 0.213 | 0.229 | 0.807 |
| DINCO | **0.088** | **0.159** | **0.865** | 0.173 | 0.182 | **0.825** | **0.058** | **0.152** | **0.859** | **0.106** | **0.190** | **0.815** |
| *NLI model:* `microsoft/deberta-large-mnli` *(406M parameters)* | | | | | | | | | | | | |
| SC (Xiong et al., 2024) | 0.229 | 0.242 | 0.783 | 0.228 | 0.225 | 0.782 | 0.067 | 0.180 | 0.801 | 0.300 | 0.303 | 0.713 |
| NVC | 0.141 | 0.169 | 0.873 | **0.085** | **0.167** | 0.802 | 0.168 | 0.191 | 0.844 | 0.191 | 0.218 | 0.813 |
| DINCO | **0.103** | **0.155** | **0.878** | 0.178 | 0.181 | **0.831** | **0.045** | **0.148** | **0.864** | **0.121** | **0.190** | **0.818** |

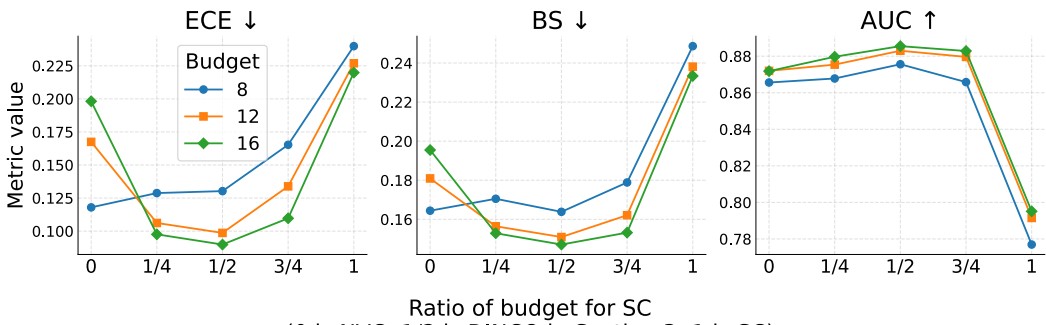

Figure 8: Calibration at different budget splits (Qwen3-8B on TriviaQA). Setting DINCO's (SC sample, distractor) split to the straightforward choice of $(\frac{1}{2}, \frac{1}{2})$ as in Section 3 appears to result in the best performance. Nevertheless, DINCO is robust to the exact budget split, achieving strong calibration across the wide range of budget splits from $(\frac{1}{4}, \frac{3}{4})$ to $(\frac{3}{4}, \frac{1}{4})$. However, a budget split of $(0, 1)$ or $(1, 0)$ with only distractors or only SC samples leads to suboptimal calibration, which is not remedied by scaling up the budget due to diminishing returns, empirically supporting our motivation (Section 2.3) for combining the confidence signals from generation and validation.

can improve performance. Fig. 8 shows that the budget split can be varied substantially while maintaining performance, revealing DINCO's robustness to the budget split hyperparameters. Thus, we recommend the equal budget split for simplicity. However, we find that regardless of the budget, using no SC samples or no distractors performs worse than allocating at least part of the budget to each component, emphasizing the benefit of jointly scaling the inference budgets for generation and validation.

## D.5    OPEN-SOURCE LLM SIZE

Table 1 shows that DINCO achieves strong calibration across small to medium-scale (2-8B) open-source models, and Table 2 extends this result to frontier models. Here we consider scaling up within open models, which helps more precisely characterize DINCO's performance across scales due to

Table 13: TriviaQA results with Qwen3-32B. Text styling follows Table 1. The effectiveness of DINCO extends from the 2-8B scale (Table 1) to the 32B scale.

| Method | ECE ↓ | BS ↓ | AUC ↑ |
|---|---|---|---|
| VC (Kadavath et al., 2022) | 0.153 | 0.158 | 0.862 |
| K-VC (Tian et al., 2023) | 0.186 | 0.211 | 0.737 |
| MSP (Fadeeva et al., 2023) | 0.113 | 0.164 | 0.792 |
| SC-VC (Xiong et al., 2024) | 0.166 | 0.213 | 0.707 |
| SC (Xiong et al., 2024) | 0.129 | 0.190 | 0.741 |
| NVC | 0.194 | 0.169 | **0.880** |
| DINCO | **0.065** | **0.131** | 0.863 |

known open model sizes. Table 13 confirms that DINCO continues to achieve strong calibration at the 32B model size.

## D.6 DINCO IN THE MEDICAL DOMAIN

We test the generalization of DINCO to other domains by evaluating on BioASQ (Krithara et al., 2023), a dataset reflecting the information needs of biomedical experts. We sample 1000 factoid questions from the Task B training set of the 2026 challenge edition BioASQ14; see Table 14 for examples. Given the technical expertise required for this task, we evaluate Qwen3-32B, thus also building on Appendix D.5 to confirm DINCO's effectiveness on large open-source models. Table 15 shows that the calibration of DINCO – under the same setup as with TriviaQA – generalizes to the biomedical domain.

Table 14: Example factoid questions and answers from BioASQ.

| Question | True answer |
|---|---|
| What is the typical alteration of the thyroid profile metabolism early after coronary artery bypass graft surgery? | Low T3 syndrome occurs frequently after CABG |
| What is the prognostic role of alterred thyroid profile after cardiosurgery? | Altered thyroid profile after cardiosurgery is associated with several events in adults and in children |
| Which is the most common gene signature in Rheumatoid Arthritis patients? | Interferon signature |

Table 15: BioASQ results with Qwen3-32B. Text styling follows Table 1. DINCO generalizes to a setting requiring biomedical expertise.

| Method | ECE ↓ | BS ↓ | AUC ↑ |
|---|---|---|---|
| VC (Kadavath et al., 2022) | 0.255 | 0.257 | 0.801 |
| K-VC (Tian et al., 2023) | 0.256 | 0.270 | 0.749 |
| MSP (Fadeeva et al., 2023) | 0.215 | 0.249 | 0.743 |
| SC-VC (Xiong et al., 2024) | 0.155 | 0.233 | 0.710 |
| SC (Xiong et al., 2024) | 0.120 | 0.217 | 0.727 |
| NVC | 0.107 | 0.167 | **0.828** |
| DINCO | **0.071** | **0.160** | 0.822 |

## D.7 DISTRACTOR GENERATION AND VALIDATION IN ONE STEP

Our scaling analysis in Section 4 shows that, while DINCO incurs a slightly higher cost than SC, SC cannot match the calibration of DINCO even when scaled arbitrarily. Here, we consider further optimizing efficiency by addressing the main reason for DINCO's higher cost: the separate step to

elicit a verbalized confidence for each distractor. To combine the generation and validation of a distractor into a single conversation, we add a user response containing `Output "Yes" if your answer is correct or "No" if not.` after the LLM's answer. Thus, after generating the distractor, only one more input sentence and one more output token need to be processed to obtain a verbalized confidence. We remark, however, that this approach is not applicable to API-access models for which we cannot seamlessly switch between beam search and ordinary decoding, which motivated our two-step design that can be shared between open and closed models. Furthermore, decoupling the generation and validation of distractors allows for other variants, such as using a smaller model to generate distractors for efficiency, as discussed in Appendix C.1. Nonetheless, here we investigate the potential for a one-step design to reduce cost in certain settings while maintaining performance. Table 16 shows that the one-pass variant achieves similar calibration to the two-pass variant, while reducing the relative cost over SC from 32% to 10%.

Table 16: Comparison of DINCO's original two-step design (separate calls for distractor generation and validation) and the one-step design (combining both parts into one conversation for efficiency). With Qwen3-8B on 1000 TriviaQA questions, SC uses $9.0 \times 10^{15}$ FLOPs, DINCO (2-step) uses $1.2 \times 10^{16}$ (**32%** more than SC), and DINCO (1-step) uses $9.9 \times 10^{15}$ (**10%** more than SC). The similar calibration between the two-step and one-step variants of DINCO suggests that a distractor can be generated and validated jointly to minimize overhead without compromising calibration.

| Method | Qwen3-8B | | | Qwen3-1.7B | | | Llama-3.2-3B-Instruct | | | Gemma-3-4B-IT | | |
|---|---|---|---|---|---|---|---|---|---|---|---|---|
| | ECE↓ | BS↓ | AUC↑ | ECE↓ | BS↓ | AUC↑ | ECE↓ | BS↓ | AUC↑ | ECE↓ | BS↓ | AUC↑ |
| SC (Xiong et al., 2024) | 0.236 | 0.244 | 0.785 | 0.233 | 0.229 | 0.780 | 0.065 | 0.177 | 0.808 | 0.303 | 0.304 | 0.713 |
| DINCO (2-step) | 0.097 | 0.155 | 0.879 | 0.177 | 0.179 | 0.835 | 0.044 | 0.148 | 0.864 | 0.121 | 0.191 | 0.817 |
| DINCO (1-step) | 0.109 | 0.171 | 0.852 | 0.165 | 0.191 | 0.798 | 0.051 | 0.157 | 0.851 | 0.089 | 0.210 | 0.772 |

