# OpenReview forum: "Calibrating Verbalized Confidence with Self-Generated Distractors"
_ICLR.cc/2026/Conference — ICLR 2026 Poster_

### Official Review · Reviewer_tbbM · 2025-10-27

**Soundness:** 3
**Presentation:** 2
**Contribution:** 3
**Rating:** 6
**Confidence:** 3

**Summary:**

This paper studies the overconfidence issue in LLM verbalized confidence when faced with claims that they have little knowledge about, and proposes Distractor-Normalized Coherence (DINCO) to address such suggestibility bias. The proposed approach normalizes the verbalized confidence by considering multiple self-generated distractors. Extensive experiments have shown the effectiveness of DINCO by demonstrating how it outperforms the existing baselines even when they scale up.

**Strengths:**

1. This paper provides a novel insight that the overconfidence of LLMs comes from suggestibility and shows empirical evidence.
2. Clear motivation and preliminaries are given to support the proposed method.
3. The proposed solution DINCO is effective with extensive experiments. Comparison across multiple datasets, models, and baselines has shown that DINCO generally improves calibration metrics.

**Weaknesses:**

1. Since the entire method builds upon the observation that the model will provide higher confidence on incorrect samples, it would be better if more analysis is provided. For e.g., current results are only derived from TriviaQA. Results from other datasets will make the observation more convincing.
2. Table 5 shows that accessing an NLI model significantly improves the results. However, only one NLI model has been tested in the results. This is not general enough, and more ablation studies should be provided. How sensitive are the results to the choice of the NLI model? Are there any requirements on the NLI model?
3. Sometimes the model could generate high confidence for an incorrect answer due to false training data or hallucinations. How can we distinguish suggestibility from these cases?

**Questions:**

Please see the weaknesses.

---

> ### Author Response · Authors · 2025-11-19
> **Response to Reviewer tbbM**
>
> Thank you for highlighting our “novel insight” on the phenomenon of suggestibility, our “clear motivation”, and our “extensive experiments”. We address your comments below.
>
> > Since the entire method builds upon the observation that the model will provide higher confidence on incorrect samples, it would be better if more analysis is provided. For e.g., current results are only derived from TriviaQA. Results from other datasets will make the observation more convincing.
>
> Based on your suggestion, in Appendix D.1, we have **added an extension of this analysis to SimpleQA and biography generation (FactScore), where we find the same trends** as in Section 2 of the model assigning higher confidence to claims for incorrectly answered questions, further supporting the phenomenon of suggestibility under epistemic uncertainty.
>
> > Table 5 shows that accessing an NLI model significantly improves the results. However, only one NLI model has been tested in the results. This is not general enough, and more ablation studies should be provided. How sensitive are the results to the choice of the NLI model? Are there any requirements on the NLI model?
>
> To address this question, in Appendix D.3, we have **added an ablation on the choice of the NLI model (spanning 4 models) and show that DINCO is robust to this choice**.
>
> > Sometimes the model could generate high confidence for an incorrect answer due to false training data or hallucinations. How can we distinguish suggestibility from these cases?
>
> While it is possible that an LLM is epistemically *certain* about a false statement due to false training data, it is much more common that false training data is mixed in with true training data or that there is little relevant training data, both of which lead to epistemic uncertainty, which we find in turn leads to suggestibility.
>
> Hallucination is similar to suggestibility in that it often arises from epistemic uncertainty (Kalai et al., 2025), but they occupy different ends of the generation-validation spectrum. While hallucination occurs at the generation stage, the phenomenon of suggestibility that we identify occurs when presenting the LLM with a given claim and asking it to validate the claim. Even if an LLM hallucinates a claim, we want it to be able to identify the claim as incorrect during the validation stage. DINCO leverages confidence signals in both generation (with self-consistency) and validation (with normalized verbalized confidence) to achieve strong calibration.
>
> References:
>
> Kalai et al., 2025. Why Language Models Hallucinate.

---

> ### Author Response · Authors · 2025-11-26
> **Follow-up on rebuttal**
>
> We wanted to give you a gentle reminder to see if our response addressed your comments. Based on your suggestions, **we have extended our preliminary study to SimpleQA and FactScore where we verify the same trends as in Section 2, and we have added an NLI model ablation showing DINCO’s robustness.** We hope these additional experiments address your comments and will allow you to revisit your score, and we are happy to continue engaging for the remainder of the discussion period.

---

### Official Review · Reviewer_SPC2 · 2025-10-31

**Soundness:** 3
**Presentation:** 3
**Contribution:** 3
**Rating:** 6
**Confidence:** 2

**Summary:**

The paper introduces a method called Distractor-Normalized Coherence (DINCO) to enhance the calibration of confidence estimates in LLMs. It addresses the prevalent issue of overconfidence in LLM outputs, particularly in low-accuracy claims, which can compromise user trust. By prompting LLMs to verbalize their confidence across multiple self-generated distractors, DINCO normalizes these confidence scores and mitigates biases stemming from suggestibility when the model encounters unfamiliar topics. Overall, the paper contributes a novel approach to calibrating LLM confidence, thereby improving the usability and trustworthiness of their outputs.

**Strengths:**

1. The authors provide a thorough analysis of the proposed method's performance compared to existing approaches, showcasing its advantages in both short-form and long-form generation tasks.

2. By addressing the critical issue of overconfidence in LLM outputs, the paper contributes to enhancing the trustworthiness and usability of these models in real-world applications. The findings have implications for various domains where LLMs are deployed, emphasizing the need for reliable confidence estimates to ensure user safety and informed decision-making.

**Weaknesses:**

1. In the experiments, all evaluated open-source models are relatively small. It remains unclear whether the conclusions generalize to larger models (e.g., those exceeding 30B parameters).

2. In Appendix LLM-as-a-Judge, the description of the human evaluation process is unclear. Did the authors conduct human judgments themselves, and if so, how many participants were involved?

**Questions:**

See above

---

> ### Author Response · Authors · 2025-11-19
> **Response to Reviewer SPC2**
>
> Thank you for highlighting our “thorough analysis” and the practical impact of our work on “enhancing the trustworthiness of these models in real-world applications”. We address your comments below.
>
> > In the experiments, all evaluated open-source models are relatively small. It remains unclear whether the conclusions generalize to larger models (e.g., those exceeding 30B parameters).
>
> Based on your suggestion, in Appendices D.5 and D.6, we have **added experiments with Qwen3-32B on TriviaQA and BioASQ (a dataset requiring biomedical expertise), respectively, showing that DINCO continues to perform the best at the 32B scale**. We would like to clarify that our experiments on SimpleQA in Table 2 are with frontier models (including GPT-4.1), which are presumed to exceed 30B parameters; nevertheless, the added experiment helps to more precisely characterize the generalization of DINCO to larger models.
>
> > In Appendix LLM-as-a-Judge, the description of the human evaluation process is unclear. Did the authors conduct human judgments themselves, and if so, how many participants were involved?
>
> One author of this paper conducted the human judgments. We have added this detail to Appendix B.3.

---

> ### Author Response · Authors · 2025-11-26
> **Follow-up on rebuttal**
>
> We wanted to give you a gentle reminder to see if our response addressed your comments. **We have added experiments with Qwen3-32B, showing that DINCO continues to perform the best on large open-source models.** We hope these additional experiments address your comments and will allow you to revisit your score, and we are happy to continue engaging for the remainder of the discussion period.

---

### Official Review · Reviewer_ESx7 · 2025-11-02

**Soundness:** 3
**Presentation:** 3
**Contribution:** 2
**Rating:** 4
**Confidence:** 3

**Summary:**

**Summary**: this paper proposes a method to improve the calibration of verbalized confidence approach (dubbed DiNCo), through the use of distracting contexts. Analysis across 6 models, reveals average ECE improvements of up to 0.099 over the best baseline in short-form QA datasets, and pearson correlation improvements (with passage-level FActScore) of 0.072 in long-form QA datasets (in biography generation).

**Main Contribution**:
- the proposal of a calibration method for verbalized confidence using distracting contexts, enabling more diverse and trustworthy confidence estimates.
- experiments concerning both short-form QA and long-form QA, and across various models.
- interesting analysis comparing proposed method to self-consistency baseline at different sampling budgets

**Strengths:**

1. **Relevant and interesting work in uncertainty quantification in LLMs**, focusing on improving calibration of self-confidence approaches.
2. **Novel findings** showing that it is possible to calibrate verbalized confidence estimates in a training-free manner, while alleviating confidence saturation and achieving better calibration than the best baseline irrespective of the compute budget.
3. **Empirically and theoretically grounded**: the proposed improvement follows from the concept of _suggestibility_, which states that LLMs’ generations are highly influenced by the context when LLMs lack sufficient information.
4. **Extensive experimentation including 6 models and concerning both short-form (i.e., TriviaQA, SimpleQA) and long-form QA datasets (i.e., biography generation)**.

**Weaknesses:**

1. Small improvements in calibration method, leading to questions about the significance and real impact of such improvements (e.g., one of the main results states that DiNCo improves ECE by an average of up to 0.099 (line 107)).
2. Results are difficult to generalize, due to experimental changes (e.g., methods and configurations differ from Table 2 to Table 3).
3. However, it would be nice to test generalization of the proposed approach to other domains, especially those requiring domain expertise (e.g., medical domain or legal domain).

**Questions:**

**Questions**:
1. The exact setup for the preliminary study (in Section 2.2) is slightly unclear to me. Can you please clarify how you generate the set of C claims for the TriviaQA dataset and how many claims there are? I.e., what’s the upper bound in total confidence ? Is the number of claims the same across every question?
2. The preliminary study in Section 2.2 reveals that incorrect instances are indeed more suggestible (as evidenced by the average higher total confidence). However, I’m a bit surprised that the total confidence for correct answers is not unimodal and centered at 1. Do you have any intuition on why that is?
3. Table 1 caption mentions “underline results not significantly worse” but there is no underline. Does it mean that all of the results are significantly equal or better than the bold results?
4. Line 355 claims that “On TriviaQA, DiNCo outperforms the best baseline SC, by an average of ECE OF 0.099”. Can you clarify what is the criteria for selecting SC as the best baseline? If I understand the table correctly, Table 1 shows that MSP has lower ECE than SC in 3 out of the 4 evaluated models (avg ECE: 0.187 vs 0.209, respectively), which would suggest that MSP is a better baseline in the TriviaQA dataset.

**Clarity**:
1. May be worth defining what is a valid minimal pair, since in Figure 2 left, examples may differ up to 2 words. Are minimal pairs restricted to those that differ in at most 2 words?
2. It’s worth clarifying when discussing improvements over baselines, whether these concern relative or absolute improvements.

**Formatting**:
1. Is there a reason why the bins for correct and incorrect are not centered at integer numbers and have width 1? What does the count for total confidence [0, 0.33] mean in this context? Additionally, I recommend adding the corresponding means as vertical lines to ease comparison.

---

> ### Author Response · Authors · 2025-11-19
> **Response to Reviewer ESx7 (Part 1)**
>
> Thank you for highlighting the relevance of the problem, our “novel findings”, our “empirically and theoretically grounded” method, and our “extensive experimentation”. We address your comments below.
>
> > Small improvements in calibration method, leading to questions about the significance and real impact of such improvements (e.g., one of the main results states that DiNCo improves ECE by an average of up to 0.099 (line 107)).
>
> We would like to clarify that the calibration metrics (ECE, BS, AUC) are on a scale from 0 to 1, just like accuracy, so even an improvement of 0.05 is substantial, as it corresponds to a 5% absolute improvement. We have added this clarification in Section 3.1. For example, Tian et al. (2023), in a highly influential paper on zero-source calibration, show that their method improves ECE by 0.04 and 0.09 on Claude-1 and Claude-2 (Tables 3 and 4), respectively. Xiong et al. (2024), in another highly influential paper on zero-resource calibration, show similar degrees of ECE differences between the top methods in Figure 3. Thus, we believe that the calibration gains obtained by DINCO (ECE improvements over the best baseline by an average of 0.077, 0.092, and 0.055 on TriviaQA, SimpleQA, and FactScore, respectively) should be considered substantial.
>
> > Results are difficult to generalize, due to experimental changes (e.g., methods and configurations differ from Table 2 to Table 3).
>
> Because long-form generation involves generating many tokens/claims, most methods for uncertainty quantification (UQ) in long-form generation must introduce some methodological component to obtain behavior that is not already present in UQ methods that work in multiple-choice or short-form QA (Manakul et al., 2023; Zhang et al., 2024; Zhang et al., 2025). Indeed, our SC baseline in the long-form setting (Table 3) requires substantial improvement/modification from the short-form setting (Table 2), following Zhang et al. (2024). Specifically, the SC baseline samples several biographies and prompts the LLM to determine whether each sampled biography entails each claim extracted from the main biography. See the related work section for more discussion on UQ for long-form generation. Thus, we believe that using different prompts to generate distractors in the short-form and long-form settings represents a reasonable degree of adjustment to the method. DINCO’s performance is also more robust across settings: the best baseline in short-form settings is MSP, but it is not applicable to long-form settings, where SC becomes the standard baseline; in either case, DINCO outperforms the best baseline. We would also like to clarify that the usage of FactScore for claim decomposition in the long-form setting is part of the experimental setup rather than our method, as it is a standard framework for evaluating long-form factuality (OpenAI, 2025).
>
> > However, it would be nice to test generalization of the proposed approach to other domains, especially those requiring domain expertise (e.g., medical domain or legal domain).
>
> Thanks for this suggestion – following your example of a “medical domain”, in Appendix D.6, **we have added a new experiment testing Qwen3-32B on BioASQ**, a dataset requiring biomedical expertise. We show that the calibration of DINCO – under the same setup as with TriviaQA – generalizes to the biomedical domain.
>
> > The exact setup for the preliminary study (in Section 2.2) is slightly unclear to me. Can you please clarify how you generate the set of C claims for the TriviaQA dataset and how many claims there are? I.e., what’s the upper bound in total confidence ? Is the number of claims the same across every question?
>
> We have added more explanation of the setup in Section 2.2. The setup is the same as in our experiments in Section 3. We generate distractors using the method in Section 2.3 (beam search on a vanilla QA prompt), and we compute total confidence according to Equation 4. For each question, we generate a main answer and 9 distractors, so the maximum possible total confidence is 10.

---

> ### Author Response · Authors · 2025-11-19
> **Response to Reviewer ESx7 (Part 2)**
>
> > Is there a reason why the bins for correct and incorrect are not centered at integer numbers and have width 1? What does the count for total confidence [0, 0.33] mean in this context? Additionally, I recommend adding the corresponding means as vertical lines to ease comparison.
>
> The total confidence we plot is computed according to Equation 4. Since each confidence is a number from 0 to 1, the total confidence can be any number from 0 to the number of claims. The height of the bar over [0, 0.33] thus represents the number of questions where the total confidence is in that range.
>
> Based on your helpful suggestion, we have now labeled the means and plotted the CDFs in Figure 3, as well as in Figure 7, which extends the preliminary study on TriviaQA to SimpleQA and FactScore.
>
> > The preliminary study in Section 2.2 reveals that incorrect instances are indeed more suggestible (as evidenced by the average higher total confidence). However, I’m a bit surprised that the total confidence for correct answers is not unimodal and centered at 1. Do you have any intuition on why that is?
>
> In the preliminary analysis, we take correctness as a proxy for epistemic certainty, i.e. we treat correct instances as epistemically certain and incorrect instances as epistemically uncertain. However, this is merely an approximation as it is in general highly nontrivial to define or quantify the model’s true epistemic uncertainty. The model may still be epistemically uncertain about questions it got correct. As an extreme example, if the model has a (perfectly calibrated) confidence of 50% on each of 100 questions, then about 50 of them will end up being correct, but the model was in fact equally uncertain about its correct answers as about its incorrect answers. In other words, whether the answer happened to be correct is a noisy signal of the calibrated confidence of it being correct. Despite this noise, our preliminary study identifies the clear trend that the model tends to report higher confidence on claims for incorrectly answered questions, which the model should typically be more epistemically uncertain on. We have added discussion on this to Section 2.2.
>
> > Table 1 caption mentions “underline results not significantly worse” but there is no underline. Does it mean that all of the results are significantly equal or better than the bold results?
>
> It means that the bolded result is statistically significantly better than all the other methods. We keep this description in the caption of Table 1 despite the absence of underlines because other tables refer to Table 1 for the description of text styling.
>
> > Line 355 claims that “On TriviaQA, DiNCo outperforms the best baseline SC, by an average of ECE OF 0.099”. Can you clarify what is the criteria for selecting SC as the best baseline? If I understand the table correctly, Table 1 shows that MSP has lower ECE than SC in 3 out of the 4 evaluated models (avg ECE: 0.187 vs 0.209, respectively), which would suggest that MSP is a better baseline in the TriviaQA dataset.
>
> Thank you for pointing this out! We have corrected this in Sections 1 and 3.2 (lines 107 and 368). We would also like to emphasize that although MSP does well on short-form QA, its reliance on answers having a canonical form prevents its generalization to long-form settings.
>
> > May be worth defining what is a valid minimal pair, since in Figure 2 left, examples may differ up to 2 words. Are minimal pairs restricted to those that differ in at most 2 words?
>
> We have added a description of what a minimal pair is (i.e. a statement that is similar to the main claim but that likely contradicts the main claim) in the caption of Figure 2 to address this confusion. Although the example distractors in Figure 2 differ by 2 words from the main claim, we do not prescribe a precise form for generating distractors. In Appendix C.1, we have added more examples, analysis, and discussion.
>
> > It’s worth clarifying when discussing improvements over baselines, whether these concern relative or absolute improvements.
>
> We have added a clarification that we report absolute improvements in Section 3.1.
>
> References:
>
> Manakul et al., 2023. SelfCheckGPT: Zero-Resource Black-Box Hallucination Detection for Generative Large Language Models.
>
> Zhang et al., 2024. LUQ: Long-text Uncertainty Quantification for LLMs.
>
> Zhang et al., 2025. Reinforcement Learning for Better Verbalized Confidence in Long-Form Generation.
>
> OpenAI, 2025. GPT-5 System Card.
>
> Tian et al., 2023. Just Ask for Calibration: Strategies for Eliciting Calibrated Confidence Scores from Language Models Fine-Tuned with Human Feedback.
>
> Xiong et al., 2024. Can LLMs Express Their Uncertainty? An Empirical Evaluation of Confidence Elicitation in LLMs.

---

> ### Author Response · Authors · 2025-11-26
> **Follow-up on rebuttal**
>
> We wanted to give you a gentle reminder to see if our response addressed your comments. **In addition to improving clarity based on your suggestions, we have added experiments on BioASQ to show DINCO’s generalization to domains requiring domain expertise.** We hope these additional experiments address your comments and will allow you to revisit your score, and we are happy to continue engaging for the remainder of the discussion period.

---

> > ### Comment · Reviewer_ESx7 · 2025-11-27
> >
> > I appreciate the authors taking the time and effort into clarifying and running additional experiments on the generalization to the BioASQ. I am happy with the authors’ response and have no further questions. I believe it is a good paper, so I’ll increase my score.

---

### Official Review · Reviewer_wbh4 · 2025-11-10

**Soundness:** 3
**Presentation:** 3
**Contribution:** 3
**Rating:** 4
**Confidence:** 3

**Summary:**

The paper introduces Distractor-Normalized Coherence (DINCO), a method for calibrating verbalized confidence in LLMs. The authors argue that standard verbalized confidence is often overconfident and saturated due to LLM suggestibility, that is, the tendency to assign high confidence to claims in the context, especially when the model is epistemically uncertain. The paper presents empirical results that suggest that DINCO leads to an improvement in calibration compared to baselines.

**Strengths:**

The overall idea of the proposed method is creative and explores a new direction for uncertainty quantification in LLMs. I see the main strengths of the paper as follows:

- originality: to the best of my knowledge, the connection between overconfidence and "suggestibility" is new, well-justified idea and appear to be empirically supported by the empirical results
- empirical results: DINCO appears to consistently outperform strong baselines across diverse tasks (both short-form and long-form) and models (open and closed weight) and that scaling up self-consistency to 100 samples still doesn't match DINCO (at 10 inference calls), which is compelling
- clarity: the paper is very well-written and clear

**Weaknesses:**

I believe there are a few weaknesses:

- heavy reliance on oracles
- efficiency: the paper positions DINCO as a "zero-resource" method, but the method requires multiple inference passes to generate distractors, multiple passes to validate each one, and calls to an NLI model, which is more expensive than standard SC
- distractor quality: the method assumes the LLM can generate high-quality, "minimal pair" distractors, but if a model is epistemically uncertain, it might fail to generate plausible alternatives, breaking the normalization assumption; beam search or prompting is used, but the paper doesn't rigorously analyze the quality or diversity of these distractors and how they directly impact the $\beta(C)$ factor
- hyperparameters: the choice of $K=5$ for distractors and $K=5$ for SC samples seems arbitrary. Is the method sensitive to these?

**Questions:**

- Is the proposed method sensitive to hyperparameters, especially to the choice of $K=5$ for distractors and $K=5$ for SC samples?
- Can you provide a cost analysis (latency/FLOPs) comparing DINCO to SC@10 and SC@100? The claim that DINCO is more efficient than scaling SC needs concrete validation given the multi-stage pipeline (generate distractors -> validate each -> NLI pairs -> SC samples).
- What happens when the distractor generation fails to produce mutually exclusive claims, even after NLI filtering?
- Have you evaluated how the choice of NLI model impacts downstream calibration?

---

> ### Author Response · Authors · 2025-11-19
> **Response to Reviewer wbh4 (Part 1)**
>
> Thank you for highlighting our “new, well-justified idea”, our results showing “DINCO appears to consistently outperform strong baselines”, and the clarity of our paper. We address your comments below.
>
> > heavy reliance on oracles
>
> We would like to clarify that DINCO does not rely on oracles. Could you clarify which oracles you are referring to?
>
> Specifically, DINCO uses the generator LLM to produce its own distractors, not requiring other LLMs or any ground-truth knowledge. Regarding the ability of the LLM to produce distractors, please see our response to the next comment.
>
> Moreover, DINCO uses an NLI model to downweight redundant distractors. As we mention in footnote 3, we believe that access to an NLI model poses only a minimal departure from the zero-resource setting, since NLI is a generic and largely solved task for which there are lightweight off-the-shelf models. Furthermore, our SC baseline also uses an NLI model to determine semantic equivalence when grouping together answers, following Kuhn et al. (2023); we have added this clarification in Section 3.
>
> > distractor quality: the method assumes the LLM can generate high-quality, "minimal pair" distractors, but if a model is epistemically uncertain, it might fail to generate plausible alternatives, breaking the normalization assumption; beam search or prompting is used, but the paper doesn't rigorously analyze the quality or diversity of these distractors and how they directly impact the \beta(C) factor
>
> Based on this suggestion, we have added examples, quantitative analysis, and discussion of the distractors in Appendix C.1. We consistently observe high average values of verbalized confidence, $w_{\rm unique}$, and $w_{\rm contra}$ (which underlie the computation of $\beta_C$), suggesting that many self-generated distractors fulfill the desiderata of being plausible, unique, and counterfactual. In Lines 1182-1217, we discuss your question about the model’s ability to generate good distractors when epistemically uncertain, which is something we thought about too. To summarize the discussion, since the distractors are generated as candidate answers by the validator LLM itself, we would expect it to find those candidate answers to be plausible. This is supported empirically by the high values of verbalized confidence in Figures 5 and 6.
>
> > What happens when the distractor generation fails to produce mutually exclusive claims, even after NLI filtering?
>
> We would like to clarify that we do not filter distractors in a hard sense but rather weight them with the continuous NLI-based weights $w_{\rm unique}$ and $w_{\rm contra}$. We added more explanation of the role of the NLI model in Section 2.3. Thus, the NLI model should be able to account for varying degrees of entailment and contradiction.
>
> While in general it may be difficult to identify entailment and contradiction relationships, such as in cases requiring domain expertise, we argue that NLI is a relatively simple task, and in the worst case, more powerful NLI models (or even the generator LLM itself or the human user seeking the LLM’s confidence) can be used. In Appendix D.6, we have added an experiment testing DINCO on BioASQ, showing that the same NLI model and setup used for TriviaQA generalizes to a setting requiring biomedical expertise.
>
> > Can you provide a cost analysis (latency/FLOPs) comparing DINCO to SC@10 and SC@100? The claim that DINCO is more efficient than scaling SC needs concrete validation given the multi-stage pipeline (generate distractors -> validate each -> NLI pairs -> SC samples).
>
> Based on this suggestion, we have added Figure 4 with FLOP counts (computed by multiplying the number of tokens by the FLOPs associated with linear layers) for DINCO@10, SC@10, and SC@100. **While DINCO@10 is 32% more expensive than SC@10, DINCO@10 outperforms SC@100 despite costing 7.6 times less, showing that compute is better spent on DINCO.**
>
> We note that the cost of the lightweight NLI model (184M parameters) in DINCO and SC is negligible, making up less than 1% of the total FLOP count. The slightly higher cost of DINCO is mainly due to the separate step to elicit a verbalized confidence for each distractor. In Appendix D.7, we consider combining distractor generation and validation into one step to reduce overhead, and show similar calibration to the original two-step method while reducing the relative cost over SC from 32% to 10%. In Appendix C.1, we have added a new discussion on the feasibility of generating distractors using a smaller model to reduce cost below even that of SC.

---

> ### Author Response · Authors · 2025-11-19
> **Response to Reviewer wbh4 (Part 2)**
>
> > hyperparameters: the choice of K=5 for distractors and K=5 for SC samples seems arbitrary. Is the method sensitive to these?
>
> We have added Figure 8, where we plot performance at different budgets and allocation ratios. **We find that DINCO is robust to the allocation ratio, performing similarly on (1/2 for SC samples, 1/2 for distractors), (1/4, 3/4), and (3/4, 1/4).** However, allocating the entire budget to only SC samples or only distractors leads to suboptimal calibration, emphasizing the benefit of jointly scaling the inference budgets for generation and validation. Thus, we recommend the equal budget split as in our main experiments for simplicity.
>
> > Have you evaluated how the choice of NLI model impacts downstream calibration?
>
> In Appendix D.3, **we have added an ablation of the choice of the NLI model (spanning 4 models) and show that DINCO is robust to this choice**.
>
> References:
>
> Kuhn et al., 2023. Semantic Uncertainty: Linguistic Invariances for Uncertainty Estimation in Natural Language Generation.

---

> ### Author Response · Authors · 2025-11-26
> **Follow-up on rebuttal**
>
> We wanted to give you a gentle reminder to see if our response addressed your comments. To summarize our response, **(1) we added quantitative analysis of distractors showing they are often plausible, unique, and counterfactual, (2) we added a FLOPs analysis to validate the efficiency of DINCO, (3) we added a budget hyperparameter ablation showing DINCO’s robustness, (4) we added an NLI model ablation showing DINCO’s robustness.** We hope these additional experiments address your comments and will allow you to revisit your score, and we are happy to continue engaging for the remainder of the discussion period.

---

### Author Response · Authors · 2025-11-19
**General response to reviewers**

We would like to thank the reviewers for their attentive and helpful reviews, highlighting the **novelty of our method** (Reviewers wbh4, ESx7, SPC2, tbbM), our **clear theoretical motivation** (Reviewers wbh4, ESx7, tbbM), our **extensive experiments** (Reviewers wbh4, ESx7, SPC2, tbbM), and the **practical relevance** of our work on LLM calibration (Reviewers ESx7, SPC2). Here in the general response, we summarize our major updates. We have also uploaded an updated version of our paper, which we reference in our responses, with changes and additions highlighted in blue.

- (wbh4, tbbM) In Appendix D.3, we have added an **ablation on the choice of the NLI model** (spanning 4 models) and show that DINCO is robust to this choice.
- (wbh4) We have added a **FLOPs analysis** to Figure 4, showing that although DINCO@10 is 32% more expensive than SC@10, DINCO@10 outperforms SC@100 despite costing 7.6 times less, showing that compute is better spent on DINCO.
- (wbh4) We have added Figure 8, where we plot performance at **different inference budgets and allocation ratios** (between SC samples and distractors), finding that DINCO is robust to the allocation ratio.
- (ESx7) In Appendix D.6, we have **added experiments on BioASQ**, a dataset requiring biomedical expertise. Here, we obtain similar gains, showing the domain generalization of DINCO.
- (SPC2) In Appendices D.5 and D.6, we have **added experiments with Qwen3-32B** on TriviaQA and BioASQ, respectively, showing that DINCO’s effectiveness extends to large open-source models.
- (tbbM) In Appendix D.1, we have **added an extension of our preliminary study to SimpleQA and biography generation (FactScore)**, where we find the same trends as in Section 2 of the model assigning higher confidence to claims for incorrectly answered questions, further supporting the phenomenon of suggestibility under epistemic uncertainty.
- (wbh4, ESx7) In Appendix C.1, we have **added examples, analysis, and discussion of our self-generated distractors**. Our analysis shows that our distractor generation methods enable diverse exploration of alternative claims.

---

> ### Author Response · Authors · 2025-12-02
> **Summary of rebuttal**
>
> We would like to summarize the rebuttal discussion for the convenience of the AC.
>
> Reviewer ESx7 commented that they are “happy with the authors’ response and have no further questions” and increased their score from **4 to 8** on November 27\. While the other reviewers did not have a chance to respond to our rebuttal, we believe that our additional experiments and discussions would have addressed their comments.
>
> ---
>
> In addition to our new experiments, below we summarize our discussions and clarifications to the reviewers’ main comments:
>
> > *(Reviewer wbh4)* heavy reliance on oracles
>
> We believe this is based on a misunderstanding – our method, DINCO, does not rely on oracles. We use the generator LLM to produce its own distractors, and we use an off-the-shelf NLI model following past work in zero-resource calibration (Kuhn et al., 2023).
>
> > *(Reviewer ESx7)* Small improvements in calibration method, leading to questions about the significance and real impact of such improvements (e.g., one of the main results states that DiNCo improves ECE by an average of up to 0.099 (line 107)).
>
> We clarify that the calibration metrics (ECE, BS, AUC) are on a scale from 0 to 1, just like accuracy, so even an improvement of 0.05 is substantial, as it corresponds to **a 5% absolute improvement**. We cite major papers in zero-resource calibration with similar degrees of improvement (Tian et al., 2023; Xiong et al., 2024), showing that DINCO’s calibration gains (ECE improvements over the best baseline by an average of 0.077, 0.092, and 0.055 on TriviaQA, SimpleQA, and FactScore, respectively) should be considered substantial.
>
> > *(Reviewer ESx7)* Results are difficult to generalize, due to experimental changes (e.g., methods and configurations differ from Table 2 to Table 3).
>
> Because long-form generation involves generating many tokens/claims, most methods for uncertainty quantification (UQ) in long-form generation (including our SC baseline) must introduce some methodological component to obtain behavior that is not already present in UQ methods that work in multiple-choice or short-form QA (Manakul et al., 2023; Zhang et al., 2024; Zhang et al., 2025). Thus, we believe that using different prompts to generate distractors in the short-form and long-form settings represents a reasonable degree of adjustment to our method.
>
> ---
>
> In addition to these major comments, we address the remaining minor comments and questions in the individual responses below.

---

### Meta-Review · Area_Chair_K5PC · 2026-01-11

**Summary:**

This paper proposes a novel approach to calibrate the verbalized confidence generated by LLMs, which demonstrates good performance and generalizability.

The author rebuttal has adequately addressed the major concerns raised by the reviewers. Among these, the two issues I find most important are efficiency and generalizability. While the proposed approach incurs additional computational cost, it remains within an acceptable range. Moreover, the newly added experiments address the generalizability concerns.

Overall, I would like to recommend acceptance.

**Reviewer Concerns:**

No major concerns remain.

**Reviewer Scores:**

The two reviewers giving 4 scores would have increased their scores.

---

### Decision · Program_Chairs · 2026-01-26

Accept (Poster)